# Learning Backpropagation-Free Deep Architectures with Kernels

## Abstract

One can substitute each neuron in any neural network with a kernel machine and obtain a counterpart powered by kernel machines. The new network inherits the expressive power and architecture of the original but works in a more intuitive way since each node enjoys the simple interpretation as a hyperplane (in a reproducing kernel Hilbert space). Further, using the kernel multilayer perceptron as an example, we prove that in classification and under certain losses, an optimal representation that minimizes the risk of the network can be characterized for each hidden layer. This result removes the need of backpropagation in learning the model and can be generalized to any feedforward kernel network. Moreover, unlike backpropagation, which turns models into black boxes, the optimal hidden representation enjoys an intuitive geometric interpretation, making the dynamics of learning in a deep kernel network transparent. Empirical results are provided to complement our theory.

## 1 Introduction

Any neural network (NN) can be turned into a kernel network (KN) by replacing each artificial neuron (McCulloch & Pitts, 1943), i.e., learning machine of the form $f(x) = \sigma(w^\top x + b)$, with a kernel machine, i.e., learning machine of the form $f(x) = \langle w, \phi(x) \rangle + b$ with kernel function $k(x, y) = \langle \phi(x), \phi(y) \rangle$. This combination of connectionism and kernel method enables the learning of hierarchical, distributed representations with kernels.

In terms of training, similar to NN, KN can be trained with backpropagation (BP) (Rumelhart et al., 1986). In the context of supervised learning, the need for BP in learning a deep architecture is caused by the fact that there is no explicit target information to tune the hidden layers (Rumelhart et al., 1986). Moreover, BP is usually computationally intensive and can suffer from vanishing gradient. And most importantly, BP results in hidden representations that are notoriously difficult to interpret or assess, turning deep architectures into "black boxes".

The main theoretical contribution of this paper is the following: Employing the simplest feedforward, fully-connected KN as an example, we prove that in classification and under certain losses, the optimal representation for each hidden layer that minimizes the risk of the network can be explicitly characterized. This result removes the need for BP and makes it possible to train the network in a feedforward, layer-wise fashion. And the same idea can be generalized to other feedforward KNs.

The layer-wise learning algorithm gives the same optimality guarantee as BP in the sense that it minimizes the risk. But the former is much faster and evidently less susceptible to vanishing gradient. Moreover, the quality of learning in the hidden layers can be directly assessed during or after training, providing more information about the model to the user. For practitioners, this enables completely new model selection paradigms. For example, the bad performance of the network can now be traced to a certain layer, allowing the user to debug the layers individually. Most importantly, the optimal representation for each hidden layer enjoys an intuitive geometric interpretation, making the learning dynamics in a deep KN more transparent than that in a deep NN. A simple acceleration method that utilizes the "sparse" nature of the optimal hidden representations is proposed to further reduce computational complexity.

Empirical results on several computer vision benchmarks are provided to demonstrate the competence of the model and the effectiveness of the greedy learning algorithm.

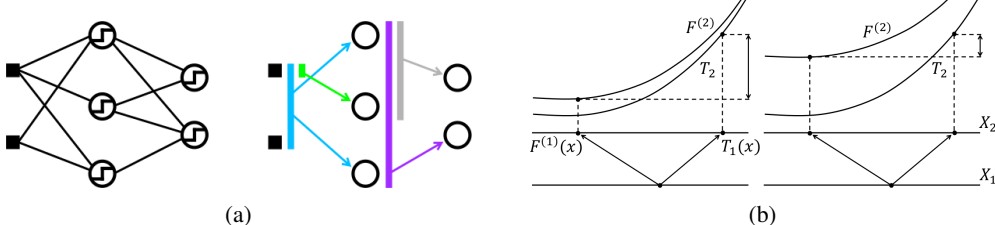

$$(a) \qquad\qquad (b)$$

Figure 1: (a) Any NN (left, presented in the usual weight-nonlinearity abstraction) can be abstracted as a "graph" (right) with each node representing a neuron and each edge the input-output relationship between neurons. If a node receives multiple inputs, we view its input as a vector in some Euclidean space, as indicated by the colored rectangles. Under this abstraction, each neuron ($f(x) = \sigma(w^\top x + b)$) can be directly replaced by a kernel machine ($f(x) = \langle w, \phi(x) \rangle + b$ with kernel $k(x, y) = \langle \phi(x), \phi(y) \rangle$) mapping from the same Euclidean space into the real line without altering the architecture and functionality of the model.

(b) Illustration for layer-wise optimality drifting away from network-optimality. Consider a two-layer network and let $T_1, T_2$ be the target function of the first and second layer, respectively. If the first layer creates error, which is illustrated by $F^{(1)}(x)$ being far away from $T_1(x)$, the composed solution $F^{(2)} \circ F^{(1)}$ on the right is better than that on the left and hence the $F^{(2)}$ on the right corresponds to the network-wise optimality of the second layer. But the $F^{(2)}$ on the left is clearly a better estimate to the layer-wise optimality $T_2$ if the quality of estimation is measured by the supremum distance.

## 2 FROM NEURAL TO KERNEL NETWORKS

In this section, we discuss how to build a KN using a given NN. First, the generic approach is described in Fig. 1a. Note that KN inherits the expressive power of the original NN since a kernel machine is a universal function approximator under mild conditions (Park & Sandberg, 1991; Micchelli et al., 2006) and the two models share the same architecture. However, KN works in a more intuitive way since each node is a simple linear model in a reproducing kernel Hilbert space (RKHS).

We now concretely define the KN equivalent of an $l$-layer Multilayer Perceptron (MLP), which we shall refer to as the kernel MLP (kMLP).[1] Given a random sample $(x_n, y_n)_{n=1}^N$, where $(x_n, y_n) \in X_1 \times Y \subset \mathbb{R}^{d_0} \times \mathbb{R}$, denote $(x_n)_{n=1}^N$ as $S$ and $(y_n)_{n=1}^N$ as $Y_S$ for convenience. For $i = 1, 2, \ldots, l$, consider kernel $k^{(i)} : X_i \times X_i \to \mathbb{R}$, $X_i \subset \mathbb{R}^{d_{i-1}}$ (for $i > 1$, $d_{i-1}$ is determined by the width of the $i - 1^{\text{th}}$ layer). $k^{(i)}(x, y) = \langle \phi^{(i)}(x), \phi^{(i)}(y) \rangle_{H_i}$, where $\phi^{(i)}$ is a mapping into RKHS $H_i$.

For $i \geq 1$, the $i^{\text{th}}$ layer in a kMLP, denoted $F^{(i)}$, is an array of $d_i$ kernel machines: $F^{(i)} : X_i \to \mathbb{R}^{d_i}$, $F^{(i)} = (f_1^{(i)}, f_2^{(i)}, \ldots, f_{d_i}^{(i)})$, a $d_i$-tuple. Let $F^{(0)}$ be the identity map on $\mathbb{R}^{d_0}$, each $f_j^{(i)} : X_i \to \mathbb{R}$ is a hyperplane in $H_i$: $f_j^{(i)}(x) = \left\langle w_j^{(i)}, \phi^{(i)}\big(F^{(i-1)} \circ \cdots \circ F^{(0)}(x)\big) \right\rangle_{H_i} + b_j^{(i)}, w_j^{(i)} \in H_i, b_j^{(i)} \in \mathbb{R}$. In practice, $f_j^{(i)}(x)$ is usually implemented as $\sum_{n=1}^N \alpha_{nj}^{(i)} k^{(i)}\big(F^{(i-1)} \circ \cdots \circ F^{(0)}(x), F^{(i-1)} \circ \cdots \circ F^{(0)}(x_n)\big) + b_j^{(i)}$, where the $\alpha_{nj}^{(i)}, b_j^{(i)} \in \mathbb{R}$ are the learnable parameters.[2] The set of mappings $\{F^{(\ell)} \circ \cdots \circ F^{(1)} : \alpha_{nj}^{(i)}, b_j^{(i)} \in \mathbb{R} \text{ for all admissible } n, j, i\}$ defines an $l$-layer kMLP. In the rest of this paper, we shall restrict our discussions to this kMLP.

## 3 ASSUMPTIONS AND NOTATIONS

We now specify the assumptions that we impose on all kernels considered in this paper. First, we consider real, continuous, symmetric kernels only and we call a kernel positive semidefinite (PSD) or positive definite (PD) if for any $S$, the kernel matrix defined as $(G)_{mn} = k(x_m, x_n)$ is PSD or PD, respectively. We shall always assume that any kernel considered is at least PSD and that

---

[1]A PyTorch-based (Paszke et al., 2017) library for implementing kMLP and the proposed layer-wise learning algorithm is available at: *anonymized URL*.

[2]The optimality of this expansion will be later justified in Section 4 under the layer-wise setting using representer theorem (Schölkopf et al., 2001).

$k^{(i)}(x, x) = c < +\infty$ for all $x \in X_i$ and $\inf_{x,y \in X_i} k^{(i)}(x, y) = a > -\infty$. It is straightforward to check using Cauchy-Schwarz inequality that the first condition implies $\max_{x,y \in X_i} k^{(i)}(x, y) = c$. For each fixed $x \in X_i$, we assume that $k^{(i)}(x, y)$, as a function of $y$, is $L_x^{(i)}$-Lipschitz with respect to the Euclidean metric on $X_i$. Let $\sup_{x \in X_i} L_x^{(i)} = L^{(i)}$, which we assume to be finite.

The following notations will be used whenever convenient: We use the shorthand $F^{(1)}(S)$ for $(F^{(1)}(x_n))_{n=1}^N$. For $i = 2, 3, \dots, l$, $F^{(i)}(S) := F^{(i)} \circ F^{(i-1)} \circ \cdots \circ F^{(1)}(S)$ and the same is true with $S$ substituted by any $x$. Throughout this paper, notations such as $F^{(i)}$ can either be used to denote a set of functions or a specific function in some set depending on the context. Also, when there is no confusion, we shall suppress the dependency of any loss function on the example for brevity, i.e., for a loss function $\ell$, instead of writing $\ell(f(x), y)$, we shall write $\ell(f)$.

# 4 A LAYER-WISE LEARNING ALGORITHM

To simplify discussion, we shall restrict ourselves to binary classification ($Y = \{+1, -1\}$) and directly give the result on classification with more than two classes in the end. A generalization to regression is left as future work. Again, we only focus on kMLP although the idea can be directly generalized to all feedforward KNs. We now discuss the layer-wise learning algorithm, beginning by addressing the difficulties with training a deep architecture layer-by-layer.

## 4.1 FUNDAMENTAL DIFFICULTIES

There are two fundamental difficulties with learning a deep architecture layer-wise. First, the hidden layers do not have supervision (labels) to learn from. And it depends on BP to propagate supervision from the output backward (Rumelhart et al., 1986). We shall prove that for kMLP, one can characterize the optimal target representation for each hidden layer, which induces a risk for that layer. The target is optimal in the sense that it minimizes the risk of the subsequent layer and eventually that of the network if all layers succeed in learning their optimal representations. This optimal representation defines what we call "layer-wise optimality".

The other difficulty with layer-wise learning is that for any given hidden layer, when the upstream layers create error, layer-wise optimality may not coincide with "network-wise optimality", i.e., the solution of this layer that eventually leads to the composed solution that minimizes the risk of the network in this suboptimal case. Indeed, when a hidden layer creates error, the objective of any layer after it becomes learning a solution that is a compromise between one that is close to the layer-wise optimality and one that prevents the error from the "bad" layer before it from "getting through" easily. And the best compromise is the network-wise optimality. The two solutions may not coincide, as shown in the toy example in Fig. 1b. Clearly, we would like to always learn the network-wise optimality at each layer, but the learner is blind to it if it is only allowed to work on one layer at a time. By decomposing the overall error of the network into error at each layer, we prove that in fact, network-wise optimality is learnable for each hidden layer even in a purely layer-wise fashion and that the proposed layer-wise algorithm learns network-wise optimality at each layer.

## 4.2 THE OPTIMAL HIDDEN REPRESENTATIONS

We now address the first difficulty in layer-wise learning. The basic idea is first described in Section 4.2.1. Then we provide technical results in Section 4.2.2 and Section 4.2.3 to fill in the details.

### 4.2.1 BASIC IDEA

Given a deep architecture $\mathbf{F} := F^{(l)} \circ \cdots \circ F^{(1)}$ and a loss function $\ell_l$ defined for this network which induces a risk that we wish to minimize: $R_l = \mathbf{E}\, \ell_l(\mathbf{F})$. BP views this problem in the following way: $R_l$ is a function of $\mathbf{F}$. The learner tries to find an $\mathbf{F}$ that minimizes $R_l$ using the random sample $S$ with labels $Y_S$ according to some learning paradigm such as Empirical Risk Minimization (ERM) or Structural Risk Minimization (SRM) (Vapnik, 2000; Shalev-Shwartz & Ben-David, 2014). $S$ is considered as fixed in the sense that it cannot be adjusted by the learner.

Alternatively, one can view $R_l$ as a function of $F^{(l)}$ and the learner tries to find an $F^{(l)}$ minimizing $R_l$ using random sample $S_{l-1}$ with labels $Y_S$ according to some learning paradigm, where $S_{l-1} := F^{(l-1)} \circ \cdots \circ F^{(1)}(S)$.[3] The advantage is that the learner has the freedom to learn both the function $F^{(l)}$ and the random sample $S_{l-1}$. And since $S_{l-1}$ determines the decision of the learning paradigm, which then determines $R_l$, $R_l$ is now essentially a function of both $F^{(l)}$ and $S_{l-1}$: $R_l = \mathbf{E}\, \ell_l(F^{(l)}, S_{l-1})$.

The key result is that independently of the actual learning of $F^{(l)}$, one can characterize the sufficient condition on $S_{l-1}$ under which $R_l$, as a function of $S_{l-1}$, is minimized, as we shall prove. In other words, the "global minimum" of $R_l$ w.r.t. $S_{l-1}$ can be explicitly identified prior to any training. This gives the optimal $S_{l-1}$, which we denote as $S_{l-1}^\star$.

Moreover, the characterization of $S_{l-1}^\star$ gives rise to a new loss function $\ell_{l-1}$ and thus also a new risk $R_{l-1}$ that is a function of $F^{(l-1)} \circ \cdots \circ F^{(1)}$. Consequently, the same reasoning would allow us to deduce $S_{l-2}^\star$ before the learner learns $F^{(l-1)}$. And this analysis can be applied to each layer, eventually leading to a greedy learning algorithm that sequentially learns $F^{(1)}, F^{(2)} \circ F^{(1)*}, \ldots, F^{(l)} \circ F^{(l-1)*} \circ \cdots \circ F^{(1)*}$, in that order, where the asterisk on the superscript indicates that the corresponding layer has been learned and frozen.

The layer-wise learning algorithm provides a framework that enjoys great flexibility. To be specific, one could stop the above analysis at any layer $i$, then learn layers $i + 1, \ldots, l$ in a greedy fashion but still learn layers $1, \ldots, i$ together with BP. Thus, it is easy to see that BP can be brought under this framework as a special case. Nevertheless, in later text, we shall stay on the one end of the spectrum where each layer is learned individually for clarity.

We now present the formal results that give the optimal hidden representations. By the reasoning above, the analysis starts from the last hidden layer (layer $l - 1$) and proceeds backward.

### 4.2.2 FORMAL RESULTS: $S_{l-1}^\star$

To begin with, we need to approximate the true classification error $R_l$ since it is not computable. To this end, we first review a well-known complexity measure.

**Definition 4.1** (*Gaussian complexity (Bartlett & Mendelson, 2002)*)*. Let $\mathcal{P}$ be a probability distribution on a metric space $X$ and suppose $x_1, \ldots, x_N$ are independent random elements distributed as $\mathcal{P}$. Let $\mathcal{F}$ be a set of functions mapping from $X$ into $\mathbb{R}$. Define*

$$\hat{\mathcal{G}}_N(\mathcal{F}) = \mathbf{E}\left[ \sup_{f \in \mathcal{F}} \left| \frac{2}{N} \sum_{n=1}^{N} g_n f(x_i) \right| \,\middle|\, x_1, \ldots, x_N \right],$$

*where $g_1, \ldots, g_N$ are independent standard normal random variables. The Gaussian complexity of $\mathcal{F}$ is $\mathcal{G}_N(\mathcal{F}) = \mathbf{E}\, \hat{\mathcal{G}}_N(\mathcal{F})$*

Intuitively, Gaussian complexity quantifies how well elements in a given function class can be correlated with a noise sequence of length $N$, i.e., the $g_n$ (Bartlett & Mendelson, 2002). Based on this complexity measure, we have the following bound on the expected classification error.

**Theorem 4.2.** *(Bartlett & Mendelson, 2002) For each $F$ mapping $S$ into $X_{l-1}$, let $\mathcal{F}_{l,A} = \Big\{ f : x \mapsto \Big\langle w, \phi^{(l)}\big(F(x)\big) \Big\rangle_{H_l} + b \,\Big|\, \|w\|_{H_l} \le A, b \in \mathbb{R} \Big\}$. Fix $A, \gamma > 0$, with probability at least $1 - \delta$ and for any $N \in \mathbb{N}$, every function $f^{(l)}$ in $\mathcal{F}_{l,A}$ satisfies*

$$P(y f^{(l)}(x) \le 0) \le \hat{R}_l(f^{(l)}) + 2\mathcal{G}_N(\mathcal{F}_{l,A}) + \left( \frac{8}{\gamma} + 1 \right) \sqrt{\frac{\log(4/\delta)}{2N}},$$

*where $\hat{R}_l(f^{(l)}) = \frac{1}{N} \sum_{n=1}^{N} \max(0, 1 - y_n f^{(l)}(x_n)/\gamma)$, the empirical hinge loss.*

*Given the assumptions on $k^{(l)}$, for any $F$, we have*

$$\mathcal{G}_N(\mathcal{F}_{l,A}) \le 2A \sqrt{\frac{c}{N}}.$$

---

[3] This is in fact a *set* of random samples as $F^{(l-1)} \circ \cdots \circ F^{(1)}$ is a *set* of functions.

Without loss of generality, we shall set hyperparameter $\gamma = 1$. We now characterize $S_{l-1}^\star$. Note that for a given $f^{(l)}$, $A = \left\|w_{f^{(l)}}\right\|_{H_l}$ is the smallest nonnegative real number such that $f^{(l)} \in \mathcal{F}_{l,A}$ and it is immediate that this gives the tightest bound in Theorem 4.2. Let $\kappa = \frac{1}{N}\sum_{n=1}^{N}\mathbf{1}_{\{y_n=+\}}$.

**Lemma 4.3** (optimal $S_{l-1}$)**.** *Given a learning paradigm minimizing $\hat{R}_l(f^{(l)}) + \tau\left\|w_{f^{(l)}}\right\|_{H_l}$ using representation $S_{l-1} = F(S)$, where $\tau$ is any positive constant satisfying $\tau < \sqrt{2(c-a)}\min(\kappa, 1-\kappa)$. Denote as $S_{l-1}^\star$ any representation satisfying*

$$k^{(l)}(F(x_+), F(x_-)) = a \quad and \quad k^{(l)}(F(x), F(x')) = c \tag{1}$$

*for all pairs of $x_+, x_-$ from distinct classes in $S$ and all pairs of $x, x'$ from the same class. Suppose the learning paradigm returns $f^{(l)\star}$ under this representation. Let $S_{l-1}^\circ$ be another representation under which the learning paradigm returns $f^{(l)\circ}$. If $f^{(l)\circ}$ achieves zero hinge loss on at least one example from each class, then for any $N \in \mathbb{N}$, $\hat{R}_l(f^{(l)\star}) + \tau\left\|w_{f^{(l)\star}}\right\|_{H_l} \leq \hat{R}_l(f^{(l)\circ}) + \tau\left\|w_{f^{(l)\circ}}\right\|_{H_l}$.*

The optimal representation $S_{l-1}^\star$, characterized by Eq. 1, enjoys a straightforward geometric interpretation: Examples from distinct classes are as distant as possible in the RKHS whereas examples from the same class are as concentrated as possible (see proof (C) of Lemma 4.3 for a rigorous justification). Intuitively, it is easy to see that such a representation is the "easiest" for the classifier.

The conditions in Eq. 1 can be concisely summarized in an ideal kernel matrix $G^\star$ defined as

$$(G^\star)_{mn} = a, \text{ if } y_m \neq y_n;$$
$$(G^\star)_{mn} = c, \text{ if } y_m = y_n.$$

And to have the $l-1^{\text{th}}$ layer learn $S_{l-1}^\star$, it suffices to train it to minimize some dissimilarity measure between $G^\star$ and the kernel matrix computed from $k^{(l)}$ and $F^{(l-1)}(S)$, which we denote $G_{l-1}$. Empirical alignment (Cristianini et al., 2002), $L^1$ and $L^2$ distances between matrices can all serve as the dissimilarity measure. To simplify discussion, we let the dissimilarity measure be the $L^1$ distance

$$\ell_{l-1}(F^{(l-1)}, (x_m, y_m), (x_n, y_n)) = |(G^\star)_{mn} - (G_{l-1})_{mn}|.$$

This specifies $\hat{R}_{l-1}(F^{(l-1)})$ as the sample mean of $(\ell_{l-1}(F^{(l-1)}, (x_m, y_m), (x_n, y_n)))_{m,n=1}^{N}$ and $R_{l-1}$ as the expectation of $\ell_{l-1}$ over $(X_1, Y) \times (X_1, Y)$. Note that due to the boundedness assumption on $k^{(l)}$, $\ell_{l-1} \leq 2\max(|c|, |a|)$.

### 4.2.3 Formal Results: $S_{l-2}^\star, \ldots, S_1^\star$

Similar to Section 4.2.2, we first need to approximate $R_{l-1}$.

**Lemma 4.4.** *For $j = 1, 2, \ldots, d_{l-1}$, let $f_j^{(l-1)} \in \mathcal{F}_{l-1}$, where $\mathcal{F}_{l-1}$ is a given hypothesis class. There exists an absolute constant $C > 0$ such that for any $N \in \mathbb{N}$, with probability at least $1 - \delta$,*

$$R_{l-1}(F^{(l-1)}) \leq \hat{R}_{l-1}(F^{(l-1)}) + \frac{4L^{(l)}Cd_{l-1}}{\max(|c|, |a|)}\mathcal{G}_N(\mathcal{F}_{l-1}) + \sqrt{\frac{8\log(2/\delta)}{N}}.$$

We are now in a position to characterize $S_{l-2}^\star$. For the following lemma only, we further assume that $k^{(l-1)}(x, y)$, as a function of $(x, y)$, depends only on and strictly decreases in $\|x-y\|_2$ for all $x, y \in X_{l-1}$ with $k^{(l-1)}(x, y) > a$, and that the infimum $\inf_{x,y \in X_{l-1}} k^{(l-1)}(x, y) = a$ is attained in $X_{l-1}$ at all $x, y$ with $\|x-y\|_2 \geq \eta$. Also assume that $\inf_{x,y \in X_{l-1}; \|x-y\|_2 < \eta}\left|\partial k^{(l-1)}(x, y)/\partial\|x-y\|_2\right| = \iota^{(l-1)}$ is defined and is positive.

Consider an $F$ mapping $S$ into $X_{l-1}$, let $\mathcal{F}_{l-1,A} = \left\{f : x \mapsto \left\langle w, \phi^{(l-1)}\big(F(x)\big)\right\rangle_{H_{l-1}} + b \,\Big|\, \|w\|_{H_{l-1}} \leq A, b \in \mathbb{R}\right\}$. For a given $F^{(l-1)} = \left(f_1^{(l-1)}, \ldots, f_{d_{l-1}}^{(l-1)}\right)$, it is immediate that $A = \max_j\left\|w_{f_j^{(l-1)}}\right\|_{H_{l-1}}$ is the smallest nonnegative real number such that $f_j^{(l-1)} \in \mathcal{F}_{l-1,A}$ for all $j$, giving the tightest bound in Lemma 4.4 (recall the bound on $\mathcal{G}_N(\mathcal{F}_{l-1,A})$ in Theorem 4.2). Let $\psi = \sum_{m,n=1}^{N}\mathbf{1}_{\{y_m \neq y_n\}}/N^2$.

**Lemma 4.5** (*optimal $S_{l-2}$*). *Given a learning paradigm minimizing $\hat{R}_{l-1}(F^{(l-1)}) + \tau \max_j \left\| w_{f_j^{(l-1)}} \right\|_{H_{l-1}}$ using representation $S_{l-2} = F(S)$, where $\tau$ is any positive constant satisfying $\tau < \sqrt{2d_{l-1}(c-a)}\psi\iota^{(l-1)}$. Denote as $S_{l-2}^\star$ any representation satisfying*

$$k^{(l-1)}(F(x_+), F(x_-)) = a \quad and \quad k^{(l-1)}(F(x), F(x')) = c$$

*for all pairs of $x_+, x_-$ from distinct classes in $S$ and all pairs of $x, x'$ from the same class. Suppose the learning paradigm returns $F^{(l-1)\star} = \left( f_1^{(l-1)\star}, \ldots, f_{d_{l-1}}^{(l-1)\star} \right)$ under this representation. Let $S_{l-2}^\circ$ be another representation under which the learning paradigm returns $F^{(l-1)\circ}$. If $F^{(l-1)\circ}$ achieves zero loss on at least one pair of examples from distinct classes, then for any $N \in \mathbb{N}$,*
$\hat{R}_{l-1}(F^{(l-1)\star}) + \tau \max_j \left\| w_{f_j^{(l-1)\star}} \right\|_{H_{l-1}} \leq \hat{R}_{l-1}(F^{(l-1)\circ}) + \tau \max_j \left\| w_{f_j^{(l-1)\circ}} \right\|_{H_{l-1}}.$

Applying this analysis to the rest of the hidden layers, it is evident that the $i^{\text{th}}$ layer, $i = 1, 2, ..., l-1$, should be trained to minimize the difference between $G^\star$ and the kernel matrix computed with $k^{(i+1)}$ and $F^{(i)}(S)$, denoted $G_i$. Generalizing to classification with more than two classes requires no change to the algorithm since the definition of $G^\star$ is agnostic to the number of classes involved in the classification task. Also note that the sufficiency of expanding the kernel machines of each layer on the training sample (see Section 2) for the learning objectives in Lemma 4.3 and Lemma 4.5 is trivially justified since the generalized representer theorem directly applies (Schölkopf et al., 2001).

Now since the optimal representation is consistent across layers, the dynamics of layer-wise learning in a kMLP is clear: The network maps the random sample sequentially through layers, with each layer trying to map examples from distinct classes as far as possible in the RKHS while keeping examples from the same class in a cluster as concentrated as possible. In other words, each layer learns a more separable representation of the sample. Eventually, the output layer works as a classifier on the final representation and since the representation would be "simple" after the mappings of the lower layers, the learned decision boundary would generalize better to unseen data, as suggested by the bounds above.

## 4.3 LEARNING NETWORK-WISE OPTIMALITY

We now discuss how to design a layer-wise learning algorithm that learns network-wise optimality at each layer. A rigorous description of the problem of layer-wise optimality drifting away from network-wise optimality and the search for a solution begins with the following bound on the total error of any two consecutive layers in a kMLP.

**Lemma 4.6.** *For any $i = 2, \ldots, l$, let the target function and the approximation function be $T_i, F^{(i)*} : X_i \to \mathbb{R}^{d_i}$, respectively. Let $\epsilon_i = \left\| F^{(i)*} - T_i \right\|_s := \sup_{x \in X_i} \left\| F^{(i)*}(x) - T_i(x) \right\|_2$, we have*

$$\left\| F^{(i)*} \circ F^{(i-1)*} - T_i \circ T_{i-1} \right\|_s \leq \epsilon_i + \sqrt{\epsilon_{i-1}} \sqrt{2L^{(i)} \sum_{j=1}^{d_i} \left\| w_{f_j^{(i)*}} \right\|_{H_i}^2}. \tag{2}$$

By applying the above bound sequentially from the input layer to the output, we can decompose the error of an arbitrary kMLP into the error of the layers. This result gives a formal description of the problem: The hypothesis with the minimal norm minimizes the propagated error from upstream, but evidently, this hypothesis is not necessarily close to the layer-wise optimality $T_i$.

Moreover, this bound provides the insight needed for learning network-wise optimality individually at each layer: For the $i^{\text{th}}$ layer, $i \geq 2$, searching for network-wise optimality amounts to minimizing the r.h.s. of Eq. 2. Lemma 4.3 and Lemma 4.5 characterized $T_i$ for $i < l$ and learning objectives that bound $\epsilon_i$ were provided earlier in the text accordingly. Based on those results, the solution that minimizes the new learning objective $\hat{R}_i(F^{(i)}) + \tau' \max_j \left\| w_{f_j^{(i)}} \right\|_{H_i}$, where $\tau' > 0$ is a hyperparameter, provides a good approximate to the minimizer of the r.h.s. of Eq. 2 if, of course, $\tau'$ is chosen well. Thus, taking this as the learning objective of the $i^{\text{th}}$ layer produces a layer-wise algorithm that learns network-wise optimality at this layer. Note that for BP, one usually also needs to heuristically tune the regularization coefficient for weights as a hyperparameter.

### 4.4 Accelerating the Upper Layers

There is a natural method to accelerate the upper layers (all but the input layer): The optimal representation $F(S)$ is sparse in the sense that $\phi(F(x_m)) = \phi(F(x_n))$ if $y_m = y_n$ and $\phi(F(x_m)) \neq \phi(F(x_n))$ if $y_m \neq y_n$ (see the proof (C) of Lemma 4.3). Since a kernel machine built on this representation of the given sample is a function in the RKHS that is contained in the span of the image of the sample, retaining only one example from each class would result in exactly the same hypothesis class because trivially, we have $\{\sum_{n=1}^{N} \alpha_n \phi(F(x_n)) | \alpha_n \in \mathbb{R}\} = \{\alpha_+ \phi(F(x_+)) + \alpha_- \phi(F(x_-)) | \alpha_+, \alpha_- \in \mathbb{R}\}$ for arbitrary $x_+, x_-$ in $S$ when the representation $F(S)$ is optimal. Thus, after training a given layer, depending on how close the actual kernel matrix is to the ideal one, one can (even randomly) discard a large portion of centers for kernel machines of the next layer to speed up the training of it without sacrificing performance. As we will later show in the experiments, randomly keeping a fraction of the training sample as centers for upper layers produces performance comparable to or better than that obtained with using the entire training set.

## 5 Related Works

The idea of combining connectionism with kernel method was initiated by Cho & Saul (2009). In their work, an "arc cosine" kernel was so defined as to imitate the computations performed by a one-layer MLP. Zhuang et al. (2011) extended the idea to arbitrary kernels with a focus on MKL, using an architecture similar to a two-layer kMLP. As a further generalization, Zhang et al. (2017) independently proposed kMLP and the KN equivalent of CNN. However, they did not extend the idea to any arbitrary NN. Scardapane et al. (2017) proposed to reparameterize each nonlinearity in an NN with a kernel expansion, resulting in a network similar to KN but is trained with BP. There are other works aiming at building "deep" kernels using approaches that are different in spirit from those above. Wilson et al. (2016) proposed to learn the covariance matrix of a Gaussian process using an NN in order to make the kernel "adaptive". This idea also underlies the now standard approach of combining a deep NN with an SVM for classification, which was first explored by Huang & LeCun (2006) and Tang (2013). Such an interpretation can be given to KNs as well, as we point out in Appendix B.5. Mairal et al. (2014) proposed to learn hierarchical representations by learning mappings of kernels that are invariant to irrelevant variations in images.

Much works have been done to improve or substitute BP in learning a deep architecture. Most aim at improving the classical method, working as add-ons for BP. The most notable ones are perhaps the unsupervised greedy pre-training techniques proposed by Hinton et al. (2006) and Bengio et al. (2007). Among works that try to completely substitute BP, none provided a comparable optimality guarantee in theory as that given by BP. Fahlman & Lebiere (1990) pioneered the idea of greedily learn the architecture of an NN. In their work, each new node is added to maximize the correlation between its output and the residual error signal. Several authors explored the idea of approximating error signals propagated by BP locally at each layer or each node (Bengio, 2014; Carreira-Perpinan & Wang, 2014; Lee et al., 2015; Balduzzi et al., 2015; Jaderberg et al., 2016). Kulkarni & Karande (2017) proposed to train NN layer-wise using an ideal kernel matrix that is a special case of that in our work. No theoretical results were provided to justify its optimality for NN. Zhou & Feng (2017) proposed a BP-free deep architecture based on decision trees, but the idea is very different from ours. Raghu et al. (2017) attempted to quantify the quality of hidden representations toward learning more interpretable deep architectures, sharing a motivation similar to ours.

## 6 Experiments

We compared kMLP learned using the proposed greedy algorithm with other popular deep architectures including MLP, Deep Belief Network (DBN) (Hinton & Salakhutdinov, 2006) and Stacked Autoencoder (SAE) (Vincent et al., 2010), with the last two trained using a combination of un-supervised greedy pre-training and standard BP (Hinton et al., 2006; Bengio et al., 2007). Note that we only focused on comparing with the standard, generic architectures because kMLP, as the KN equivalent of MLP, does not have a specialized architecture or features designed for specific application domains. Several optimization and training techniques were applied to the MLPs to boost performance. These include Adam (Kingma & Ba, 2014), RMSProp (Tieleman & Hinton, 2012), dropout (Srivastava et al., 2014) and batch normalization (BN) (Ioffe & Szegedy, 2015).

Table 1: Test errors (%) and 95% confidence intervals (%). When two results have overlapping confidence intervals, they are considered equivalent. Best results are marked in bold. The numbers following the model names indicate the number of hidden layers used. For kMLP$^{\text{FAST}}$, we also include (in parentheses) the portion of centers retained, i.e., the number of training examples randomly chosen as centers for the given layer divided by the size of the training set.

(a) Comparing kMLPs (trained layer-wise) with other popular deep architectures trained with BP and BP enhanced by unsupervised greedy pre-training.

| | RECTANGLES | RECTANGLES-IMAGE | CONVEX | MNIST (10K) | MNIST (10K) ROTATED | FASHION-MNIST |
|---|---|---|---|---|---|---|
| MLP-1 (SGD) | 7.16± 0.23 | 33.20± 0.41 | 32.25± 0.41 | 4.69± 0.19 | 18.11± 0.34 | 15.47± 0.71 |
| MLP-1 (ADAM) | 5.37± 0.20 | 28.82± 0.40 | 30.07± 0.40 | 4.71± 0.19 | 18.64± 0.34 | 12.98± 0.66 |
| MLP-1 (RMSPROP+BN) | 5.37± 0.20 | 23.81± 0.37 | 28.60± 0.40 | 4.57± 0.18 | 18.75± 0.34 | 14.55± 0.69 |
| MLP-1 (RMSPROP+DROPOUT) | 5.50± 0.20 | 23.67± 0.37 | 36.28± 0.42 | 4.31± 0.18 | 14.96± 0.31 | 12.86± 0.66 |
| MLP-2 (SGD) | 5.05± 0.19 | **22.77± 0.37** | 25.93± 0.38 | 5.17± 0.19 | 18.08± 0.34 | 12.94± 0.66 |
| MLP-2 (ADAM) | 4.36± 0.18 | 25.69± 0.38 | 25.68± 0.38 | 4.42± 0.18 | 17.22± 0.33 | **11.48± 0.62** |
| MLP-2 (RMSPROP+BN) | 4.22± 0.18 | 23.12± 0.37 | 23.28± 0.37 | 3.57± 0.16 | 13.73± 0.30 | **11.51± 0.63** |
| MLP-2 (RMSPROP+DROPOUT) | 4.75± 0.19 | 23.24± 0.37 | 34.73± 0.42 | 3.95± 0.17 | 13.57± 0.30 | **11.05± 0.61** |
| DBN-1 | 4.71± 0.19 | 23.69± 0.37 | 19.92± 0.35 | 3.94± 0.17 | 14.69± 0.31 | N/A |
| DBN-3 | 2.60± 0.14 | **22.50± 0.37** | **18.63± 0.34** | 3.11± 0.15 | **10.30± 0.27** | N/A |
| SAE-3 | 2.41± 0.13 | 24.05± 0.37 | **18.41± 0.34** | 3.46± 0.16 | **10.30± 0.27** | N/A |
| KMLP-1 | **2.24± 0.13** | 23.29± 0.37 | 19.15± 0.34 | **3.10± 0.15** | 11.09± 0.28 | 11.72± 0.63 |
| KMLP-1$^{\text{FAST}}$ | **2.36± 0.13 (0.05)** | 23.86± 0.37 (0.01) | 20.34± 0.35 (0.17) | **2.95± 0.15 (0.1)** | 12.61± 0.29 (0.1) | 11.45± 0.62 (0.28) |
| KMLP-2 | **2.24± 0.13** | 23.30± 0.37 | 18.53± 0.34 | 3.16± 0.15 | **10.53± 0.27** | 11.25± 0.62 |
| KMLP-2$^{\text{FAST}}$ | **2.21± 0.13 (0.3/0.3)** | 23.24± 0.37 (0.01/0.3) | 19.32± 0.35 (0.005/0.03) | 3.18± 0.15 (0.3/0.3) | **10.94± 0.27 (0.1/0.7)** | 11.15± 0.62 (1/0.28) |

(b) Further testing the proposed layer-wise learning algorithm and acceleration method using standard MNIST.

| MLP-1 (RMSPROP+BN) | MLP-1 (RMSPROP+DROPOUT) | MLP-2 (RMSPROP+BN) | MLP-2 (RMSPROP+DROPOUT) | KMLP-1 (BP) | KMLP-1 (GREEDY) | KMLP-1$^{\text{RFF}}$ (BP) |
|---|---|---|---|---|---|---|
| 2.05± 0.28 | 1.77± 0.26 | **1.58± 0.24** | 1.67± 0.25 | 3.44± 0.36 | 1.77± 0.26 | 2.01± 0.28 |

| KMLP-1$^{\text{PARAM}}$ (BP) | KMLP-1$^{\text{FAST}}$ (GREEDY) | KMLP-2 (BP) | KMLP-2 (GREEDY) | KMLP-2$^{\text{RFF}}$ (BP) | KMLP-2$^{\text{PARAM}}$ (BP) | KMLP-2$^{\text{FAST}}$ (GREEDY) |
|---|---|---|---|---|---|---|
| 1.88± 0.27 | 1.75± 0.26 (0.54) | 3.66± 0.37 | **1.56± 0.24** | 1.92± 0.27 | 2.45± 0.30 | **1.47± 0.24 (1/0.19)** |

kMLP accelerated using the proposed method (kMLP$^{\text{FAST}}$) was also compared. For these models, we randomly retained a subset of the centers of each upper layer before its training. As for the benchmarks used, *rectangles, rectangles-image* and *convex* are binary classification datasets, *mnist (10k)* and *mnist (10k) rotated* are variants of MNIST (Larochelle et al., 2007; LeCun et al., 2010). And *fashion-mnist* is the Fashion-MNIST dataset (Xiao et al., 2017). To further test the proposed layer-wise learning algorithm and the acceleration method, we compared greedily-trained kMLP with MLP and kMLP trained using BP (Zhang et al., 2017) using the standard MNIST (LeCun et al., 2010). Two popular acceleration methods for kernel machines were also compared on the same benchmark, including using a parametric representation (i.e., for each node in a kMLP, $f(x) = k(w, x)$, $w$ learnable) (kMLP$^{\text{PARAM}}$) and using random Fourier features (kMLP$^{\text{RFF}}$) (Rahimi & Recht, 2008). More details for the experiments can be found in Appendix A[4].

From Table 1a, we see that the performance of kMLP is on par with some of the most popular and most mature deep architectures. In particular, the greedily-trained kMLPs compared favorably with their direct NN equivalents, i.e., the MLPs, even though neither batch normalization nor dropout was used for the former. These results also validate our earlier theoretical results on the layer-wise learning algorithm, showing that it indeed has the potential to be a substitute for BP with an equivalent optimality guarantee. Results in Table 1b further demonstrate the effectiveness of the greedy learning scheme. For both the single-hidden-layer and the two-hidden-layer kMLPs, the layer-wise algorithm consistently outperformed BP. It is worth noting that the proposed acceleration trick, despite being extremely simple, is clearly very effective and even produced models outperforming the original ones. This shows that kMLP together with the greedy learning scheme can be of practical interest even when dealing with the massive data sets in today's machine learning.

Last but not least, we argue that it is the practical aspects that makes the greedy learning framework promising. Namely, this framework of learning makes deep architectures more transparent and intuitive, which can serve as a tentative step toward more interpretable, easy-to-understand models with strong expressive power. Also, new design paradigms are now possible under the layer-wise framework. For example, each layer can now be "debugged" individually. Moreover, since learning becomes increasingly simple for the upper layers as the representations become more and more well-behaved, these layers are usually very easy to set up and also converge very fast during training.

---

[4]Scripts for all experiments performed in this paper are available at: *anonymized URL*.

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

## APPENDIX A  EXPERIMENTAL SETUP

The first data set, known as *rectangles*, has 1000 training images, 200 validation images and 50000 test images. The learning machine is required to tell if a rectangle contained in an image has a larger width or length. The location of the rectangle is random. The border of the rectangle has pixel value 255 and pixels in the rest of an image all have value 0. The second data set, *rectangles-image*, is the same with *rectangles* except that the inside and outside of the rectangle are replaced by an image patch, respectively. *rectangles-image* has 10000 training images, 2000 validation images and 50000 test images. The third data set, *convex*, consists of images in which there are white regions (pixel value 255) on black (pixel value 0) background. The learning machine needs to distinguish if the region is convex. This data set has 6000 training images, 2000 validation images and 50000 test images. The fourth data set contains 10000 training images, 2000 validation images and 50000 test images taken from MNIST. The fifth is the same as the fourth except that the digits have been randomly rotated. Sample images from the data sets are given in Fig. 2. For actual training and testing, the pixel values were normalized to [0, 1]. For standard MNIST and Fashion-MNIST, no preprocessing was performed. For detailed descriptions of the data sets, see (Larochelle et al., 2007; LeCun et al., 2010; Xiao et al., 2017).

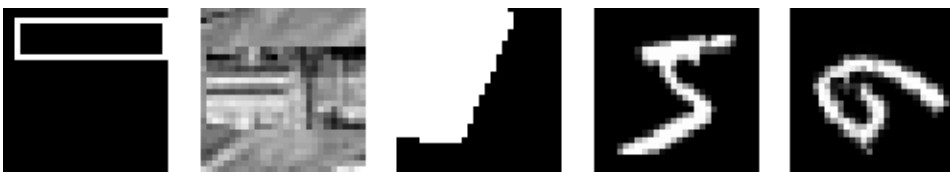

Figure 2: From left to right: example from *rectangles*, *rectangles-image*, *convex*, *mnist (10k)* and *mnist (10k) rotated*.

The experimental setup for the greedily-trained kMLPs is as follows, kMLP-1 corresponds to a one-hidden-layer kMLP with the first layer consisting of 15 to 150 kernel machines using the same Gaussian kernel ($k(x, y) = e^{-\|x-y\|^2/\sigma^2}$) and the second layer being a single or ten (depending on the number of classes) kernel machines using another Gaussian kernel. Note that the Gaussian kernel does not satisfy the condition that the infimum $a$ is attained (see the extra assumptions before Lemma 4.5), but for practical purposes, it suffices to set the corresponding entries of the ideal kernel matrix to some small value. For all of our experiments, we set $(G^\star)_{mn} = 1$ if $y_m = y_n$ and 0 otherwise. Hyperparameters were selected using the validation set. The validation set was then used in final training only for early-stopping based on validation error. For the standard MNIST and Fashion-MNIST, the last 5000 training examples were held out as validation set. For other datasets, see (Larochelle et al., 2007). kMLP-1$^{\text{FAST}}$ is the same kMLP for which we accelerated by randomly choosing a fraction of the training set as centers for the second layer after the first had been trained. The kMLP-2 and kMLP-2$^{\text{FAST}}$ are the two-hidden-layer kMLPs, the second hidden layers of which contained 15 to 150 kernel machines. We used Adam (Kingma & Ba, 2014) as the optimization algorithm of the layer-wise scheme. Although some of the theoretical results presented earlier in the paper were proved under certain losses, we did not notice a significant performance difference between using $L^1$, $L^2$ and empirical alignment as loss function for the hidden layers. And neither was such difference observed between using hinge loss and cross-entropy for the output layer. This suggests that these results may be proved in more general settings. To make a fair comparison with the NN models, the overall loss functions of all models were chosen to be the cross-entropy loss. Settings of all the kMLPs trained with BP can be found in (Zhang et al., 2017). Note that because it is extremely time/memory-consuming to train kMLP with BP without any acceleration method, to make training possible, we could only randomly use 10000 examples from the entire training set of 55000 examples as centers for the kMLP-2 (BP) from Table 1b.

We compared kMLP with a one/two-hidden-layer MLP (MLP-1/MLP-2), a one/three-hidden-layer DBN (DBN-1/DBN-3) and a three-hidden-layer SAE (SAE-3). For these models, hyperparameters were also selected using the validation set. For the MLPs, the sizes of the hidden layers were chosen from the interval [25, 700]. All hyperparameters involved in Adam, RMSProp and BN were set to the suggested default values in the corresponding papers. If used, dropout and BN was added to

each hidden layer, respectively. For DBN-3 and SAE-3, the sizes of the three hidden layers varied in intervals [500, 3000], [500, 4000] and [1000, 6000], respectively. DBN-1 used a much larger hidden layer than DBN-3 to obtain comparable performance. A simple calculation shows that the total numbers of parameters in the kMLPs were fewer than those in the corresponding DBNs and SAEs by orders of magnitude in all experiments. Like in the training for the kMLPs, the validation set were also reserved for early-stopping in final training. The DBNs and SAEs had been pre-trained unsupervisedly before the supervised training phase, following the algorithms described in (Hinton et al., 2006; Bengio et al., 2007). More detailed settings for these models were reported in (Larochelle et al., 2007).

## APPENDIX B    FURTHER ANALYSIS

In this section, we provide some further analysis on kMLP and the layer-wise learning algorithm. Namely, in Appendix B.1, we give a bound on the Gaussian complexity of an $l$-layer kMLP, which describes the intrinsic model complexity of kMLP. In particular, the bound describes the relationship between the depth/width of the model and the complexity of its hypothesis class, providing useful information for model selection. In Appendix B.2, we give a constructive result stating that the dissimilarity measure being optimized at each hidden layer will not increase as training proceeds from the input layer to the output. This also implies that a deeper kMLP performs at least as well as its shallower counterparts in minimizing any loss function they are trained on. In Appendix B.3, a result similar to Lemma 4.3 is provided, stating that the characterization for the optimal representation can be made much simpler if one uses a more restricted learning paradigm. In fact, in contrast to Lemma 4.3, both necessary and sufficient conditions can be determined under the more restricted setting. In Appendix B.4, we provide a standard, generic method to estimate the Lipschitz constant of a continuously differentiable kernel, as this quantity has been repeatedly used in many of our results in this paper. In Appendix B.5, we state some advantages of kMLP over classical kernel machines. In particular, empirical results are provided in Appendix B.5.1, in which a two-layer kMLP consistently outperforms the classical Support Vector Machine (SVM) (Cortes & Vapnik, 1995) as well as several SVMs enhanced by Multiple Kernel Learning (MKL) algorithms (Bach et al., 2004; Gönen & Alpaydın, 2011).

### B.1    GAUSSIAN COMPLEXITY OF KMLP

We first give a result on the Gaussian complexity of a two-layer kMLP.

**Lemma B.1.** *Given kernel $k : X_2 \times X_2 \to \mathbb{R}$, where $X_2 \subset \mathbb{R}^{d_1}$. Let $\mathcal{F}_2 = \{f : X_2 \to \mathbb{R}, f(x) = \sum_{\nu=1}^{m} \alpha_\nu k(x_\nu, x) + b \mid \boldsymbol{\alpha} = (\alpha_1, \dots, \alpha_m) \in \mathbb{R}^m, \|\boldsymbol{\alpha}\|_1 \leq A, b \in \mathbb{R}\}$, where the $x_\nu$ are arbitrary examples from $X_2$.*

*Consider $\mathcal{F}_1 = \{(f_1, \dots, f_{d_1}) : X_1 \to X_2 \mid f_j \in \Omega\}$, where $\Omega$ is a given hypothesis class that is closed under negation, i.e., if $f \in \Omega$, then $-f \in \Omega$. Define $\mathcal{F}_2 \circ \mathcal{F}_1 = \{h : x \mapsto \sum_{\nu=1}^{m} \alpha_\nu k(F(x_\nu), F(x)) + b \mid \|\boldsymbol{\alpha}\|_1 \leq A, b \in \mathbb{R}, F \in \mathcal{F}_1\}$.*

*If the range of some element in $\Omega$ contains 0, we have*

$$\mathcal{G}_N(\mathcal{F}_2 \circ \mathcal{F}_1) \leq A L d_1 \mathcal{G}_N(\Omega).$$

The above result can be easily generalized to kMLP with an arbitrary number of layers.

**Lemma B.2.** *Given an $l$-layer kMLP, for each $f_j^{(i)}(x) = \sum_{\nu=1}^{m} \alpha_{\nu j}^{(i)} k^{(i)}(F^{(i-1)}(x_\nu), F^{(i-1)}(x)) + b$, $\boldsymbol{\alpha}_j^{(i)} := (\alpha_{1j}^{(i)}, \dots, \alpha_{mj}^{(i)}) \in \mathbb{R}^m, b \in \mathbb{R}$, assume $\|\boldsymbol{\alpha}_j^{(i)}\|_1 \leq A_i$ and let $d_l = 1$. Denote the class of functions implemented by this kMLP as $\mathcal{F}_l$, we have*

$$\mathcal{G}_N(\mathcal{F}_l) \leq d_1 \prod_{i=2}^{l} A_i L^{(i)} d_i \mathcal{G}_N(\Omega).$$

*Proof.* It is trivial to check that the hypothesis class of each layer is closed under negation and that there exists a function in each of these hypothesis classes whose range contains 0. Then the result follows from repeatedly applying Lemma B.1. □

## B.2 Nonincreasing Loss Across Layers

**Lemma B.3.** *For $i \geq 2$, assume $k^{(i)}$ is PD and fix layers $1, 2, \ldots, i - 1$ at arbitrary states $F^{(1)}, F^{(2)}, \ldots, F^{(i-1)}$. Let the loss functions $\ell_i, \ell_{i-1}$ be the same up to their domains, and denote both $\ell_i$ and $\ell_{i-1}$ as $\ell$. Suppose layer $i$ is trained with a gradient-based algorithm to minimize the loss $\ell(F^{(i)})$. Denote the state of layer $i$ after training by $F^{(i)*}$. If $d_{i-1} = d_i$, there is an initialization for $F^{(i)}$ such that*

$$\ell(F^{(i)*}) \leq \ell(F^{(i-1)}). \tag{3}$$

*Calculation for this initialization is specified in the proof.*

*For $i = 1$, under the further assumption that $X_1 \subset \mathbb{R}^{d_0}$ and $d_0 = d_1$, Eq. 3 becomes $\ell(F^{(1)*}) \leq \ell(F^{(0)})$, where $F^{(0)}$ is the identity map on $X_1$.*

**Remark B.3.1.** *For the greedily-trained kMLP, Lemma B.3 applies to the hidden layers and implicitly requires that $k^{(i+1)} = k^{(i)}$ since the loss function for layer $i$, when viewed as a function of $F^{(i)}$, takes the form $\ell(k^{(i+1)}(F^{(i)}))$ and can be rewritten as $\ell_{k^{(i+1)}}(F^{(i)})$. Similarly, $\ell(k^{(i)}(F^{(i-1)}))$ is $\ell_{k^{(i)}}(F^{(i-1)})$. Since Lemma B.3 assumes $\ell$ to be the same across layers (otherwise it does not make sense to compare between layers), this forces $\ell_{k^{(i+1)}} = \ell_{k^{(i)}}$. Then it follows that $k^{(i+1)} = k^{(i)}$.*

*Further, if $k^{(i+1)}$ and $k^{(i)}$ have the property that $k(x, y) = k(\bar{x}, \bar{y})$, where $\bar{x}, \bar{y}$ denote the images of $x, y$ under an embedding of $\mathbb{R}^p$ into $\mathbb{R}^q$ ($p \leq q$) defined by the identity map onto a $p$-dimensional subspace of $\mathbb{R}^q$, then the condition $d_{i-1} = d_i$ can be relaxed to $d_{i-1} \leq d_i$.*

This lemma states that for a given kMLP, when it has been trained upto and including the $i^{\text{th}}$ hidden layer, the $i + 1^{\text{th}}$ hidden layer can be initialized in such a way that the value of its loss function will be lower than or equal to that of the $i^{\text{th}}$ hidden layer after training. In particular, the actual hidden representation "converges" to the optimal represetation as training proceeds across layers. On the other hand, when comparing two kMLPs, this result implies that the deeper kMLP will not perform worse in minimizing the loss function than its shallower counterpart.

In deep learning literature, results analogous to Lemma B.3 generally state that in the hypothesis class of a NN with more layers, there exists a hypothesis that approximates the target function nontrivially better than any hypothesis in that of another shallower network (Sun et al., 2016). Such an existence result for kMLP can be easily deduced from the earlier bound on its Gaussian complexity (see Lemma B.2). However, these proofs of existence do not guarantee that such a hypothesis can always be found through learning in practice, whereas Lemma B.3 is constructive in this regard. Nevertheless, one should note that Lemma B.3 does not address the risk $R = \mathbf{E}\ell$. Instead, it serves as a handy result that guarantees fast convergence of upper layers during training in practice.

## B.3 Simpler Optimal Representation under More Restricted Learning Paradigms

The following lemma states that if we are willing to settle with a more restricted learning paradigm, the necessary and sufficient condition that guarantees the optimality of a representation can be characterized and is simpler than that described in Lemma 4.3. The setup for this lemma is the same as that of Lemma 4.3 except that the assumption that the numbers of examples from the two classes are equal is not needed.

**Lemma B.4.** *Consider a learning paradigm that minimizes $\hat{R}_l(f^{(l)}) + \tau \mathcal{G}_N(\mathcal{F}_{l,A})$ using representation $S_{l-1} = F(S)$ under the constraint that it returns an $f^{(l)\star} \in \mathcal{F}_{l,A}$ with $\hat{R}_l(f^{(l)\star}) = 0$, where $\tau$ is any positive constant. For any $N \in \mathbb{N}$, $\hat{R}_l(f^{(l)\star}) + \tau \mathcal{G}_N(\mathcal{F}_{l,A})$ is minimized over all linearly separable representations if and only if the representation $F(S)$ satisfies*

$$k^{(l)}(F(x_+), F(x_-)) = a$$

*for all pairs of $x_+, x_-$ from distinct classes in $S$.*

### B.4 LIPSCHITZ CONSTANT FOR CONTINUOUSLY DIFFERENTIABLE KERNELS

In general, for a continuously differentiable function $f : \mathbb{R} \to \mathbb{R}$ with derivative $f'$ and any $a, b \in \mathbb{R}$, $a < b$, we have

$$|f(b) - f(a)| = \left| \int_a^b f'(x)dx \right| \leq \int_a^b |f'(x)|dx \leq \max_{a \leq x \leq b} |f'(x)|(b - a).$$

This simple result can be used to bound the Lipschitz constant of a continuously differentiable kernel. For example, for Gaussian kernel $k : X \times X \to \mathbb{R}$, $X \subset \mathbb{R}$, $k(x, y) = e^{-(x-y)^2/\sigma^2}$, we have $\partial k(x, y)/\partial y = 2(x - y)k(x, y)/\sigma^2$. Hence for each fixed $x \in X$, $k(x, y)$ is Lipschitz in $y$ with Lipschitz constant bounded by $\sup_{y \in X} \left| 2(x - y)k(x, y)/\sigma^2 \right|$. In practice, $X$ is always compact and can be a rather small subspace of some Euclidean space after normalization of data, hence this would provide a reasonable approximation to the Lipschitz constant of Gaussian kernel.

### B.5 COMPARING kMLP WITH CLASSICAL KERNEL MACHINES

There are mainly two issues with classical kernel machines besides their usually high computational complexity. First, despite the fact that under mild conditions, they are capable of universal function approximation (Park & Sandberg, 1991; Micchelli et al., 2006) and that they enjoy a very solid mathematical foundation (Aronszajn, 1950), kernel machines are unable to learn multiple levels of distributed representations (Bengio et al., 2013), yet learning representations of this nature is considered to be crucial for complicated artificial intelligence (AI) tasks such as computer vision, natural language processing, etc. (Bengio, 2009; LeCun et al., 2015). Second, in practice, performance of a kernel machine is usually highly dependent on the choice of kernel since it governs the quality of the accessible hypothesis class. But few rules or good heuristics exist for this topic due to its extremely task-dependent nature. Existing solutions such as MKL (Bach et al., 2004; Gönen & Alpaydın, 2011) view the task of learning an ideal kernel for the given problem to be separate from the problem itself, necessitating either designing an ad hoc kernel or fitting an extra trainable model on a set of generic base kernels, complicating training.

kMLP learns distributed, hierarchical representations because it inherits the architecture of MLP. To be specific, first, we see easily that the hidden activation of each layer, i.e., $F^{(i)}(x) \subset \mathbb{R}^{d_i}$, is a distributed representation (Hinton, 1984; Bengio et al., 2013). Indeed, just like in an MLP, each layer of a kMLP consists of an array of identical computing units (kernel machines) that can be activated independently. Further, since each layer in a kMLP is built on top of the previous layer in exactly the same way as how the layers are composed in an MLP, the hidden representations are hierarchical (Bengio et al., 2013).

Second, kMLP naturally combines the problem of learning an ideal kernel for a given task and the problem of learning the parameters of its kernel machines to accomplish that task. To be specific, kMLP performs nonparametric kernel learning alongside learning to perform the given task. Indeed, for kMLP, to build the network one only needs generic kernels, but each layer $F^{(i)}$ can be viewed as a part of a kernel of the form $k^{(i+1)}(F^{(i)}(x), F^{(i)}(y))$. The fact that each $F^{(i)}$ is learnable makes this kernel "adaptive", mitigating to some extent any limitation of the fixed generic kernel $k^{(i+1)}$. The training of layer $i$ makes this adaptive kernel optimal as a constituent part of layer $i + 1$ for the task the network was trained for. And it is always a valid kernel if the generic kernel $k^{(i+1)}$ is. Note that this interpretation has been given in a different context by Huang & LeCun (2006) and Bengio et al. (2013), we include it here only for completeness.

#### B.5.1 EMPIRICAL RESULTS

We now compare a single-hidden-layer kMLP using simple, generic kernels with SVMs enhanced by MKL algorithms that used significantly more kernels to demonstrate the ability of kMLP to automatically learn task-specific kernels out of standard ones. The standard SVM and seven other SVMs enhanced by popular MKL methods were compared (Zhuang et al., 2011), including the classical convex MKL (Lanckriet et al., 2004) with kernels learned using the extended level method proposed in (Xu et al., 2009) (MKL$^{\text{LEVEL}}$); MKL with $L^p$ norm regularization over kernel weights (Kloft et al., 2011) ($L^p$MKL), for which the cutting plane algorithm with second order Taylor approximation of

Table 2: Average test error (%) and standard deviation (%) from 20 runs. Results with overlapping 95% confidence intervals (not shown) are considered equivalent. Best results are marked in bold. The average ranks (calculated using average test error) are provided in the bottom row. When computing confidence intervals, due to the limited sizes of the data sets, we pooled the 20 random samples.

| | Size/Dimension | SVM | MKL$^{\text{LEVEL}}$ | $L^p$MKL | GMKL | IKL | MKM | 2LMKL | 2LMKL$^{\text{INF}}$ | kMLP-1 |
|---|---|---|---|---|---|---|---|---|---|---|
| Breast | 683/10 | 3.2± 1.0 | 3.5± 0.8 | 3.8± 0.7 | 3.0± 1.0 | 3.5± 0.7 | 2.9± 1.0 | 3.0± 1.0 | 3.1± 0.7 | **2.4± 0.7** |
| Diabetes | 768/8 | **23.3± 1.8** | 24.2± 2.5 | 27.4± 2.5 | 33.6± 2.5 | 24.0± 3.0 | 24.2± 2.5 | 23.4± 1.6 | 23.4± 1.9 | 23.2± 1.9 |
| Australian | 690/14 | 15.4± 1.4 | 15.0± 1.5 | 15.5± 1.6 | 20.0± 2.3 | 14.6± 1.2 | 14.7± 0.9 | 14.5± 1.6 | 14.3± 1.6 | **13.8± 1.7** |
| Iono | 351/33 | 7.2± 2.0 | 8.3± 1.9 | 7.4± 1.4 | 7.3± 1.8 | 6.3± 1.0 | 8.3± 2.7 | 7.7± 1.5 | 5.6± 0.9 | **5.0± 1.4** |
| Ringnorm | 400/20 | **1.5± 0.7** | 1.9± 0.8 | 3.3± 1.0 | 2.5± 1.0 | 1.5± 0.7 | 2.3± 1.0 | 2.1± 0.8 | 1.5± 0.8 | 1.5± 0.6 |
| Heart | 270/13 | 17.9± 3.0 | 17.0± 2.9 | 23.3± 3.8 | 23.0± 3.6 | 16.7± 2.1 | 17.6± 2.5 | 16.9± 2.5 | 16.4± 2.1 | **15.5± 2.7** |
| Thyroid | 140/5 | 6.1± 2.9 | 7.1± 2.9 | 6.9± 2.2 | 5.4± 2.1 | 5.2± 2.0 | 7.4± 3.0 | 6.6± 3.1 | 5.2± 2.2 | **3.8± 2.1** |
| Liver | 345/6 | 29.5± 4.1 | 37.7± 4.5 | 30.6± 2.9 | 36.4± 2.6 | 40.0± 2.9 | 29.9± 3.6 | 34.0± 3.4 | 37.3± 3.1 | **28.9± 2.9** |
| German | 1000/24 | 24.8± 1.9 | 28.6± 2.8 | 25.7± 1.4 | 29.6± 1.6 | 30.0± 1.5 | 24.3± 2.3 | 25.2± 1.8 | 25.8± 2.0 | **24.0± 1.8** |
| Waveform | 400/21 | 11.0± 1.8 | 11.8± 1.6 | 11.1± 2.0 | 11.8± 1.8 | 10.3± 2.3 | 10.0± 1.6 | 11.3± 1.9 | **9.6± 1.6** | 10.3± 1.9 |
| Banana | 400/2 | 10.3± 1.5 | **9.8± 2.0** | 12.5± 2.6 | 16.6± 2.7 | 9.8± 1.8 | 19.5± 5.3 | 13.2± 2.1 | 9.8± 1.6 | 11.5± 1.9 |
| Rank | - | 4.2 | 6.3 | 7.0 | 6.9 | 4.3 | 5.4 | 5.0 | 2.8 | 1.6 |

$L^p$ was adopted; Generalized MKL in (Varma & Babu, 2009) (GMKL), for which the target kernel class was the Hadamard product of single Gaussian kernel defined on each dimension; Infinite Kernel Learning in (Gehler & Nowozin, 2008) (IKL) with MKL$^{\text{LEVEL}}$ as the embedded optimizer for kernel weights; 2-layer Multilayer Kernel Machine in (Cho & Saul, 2009) (MKM); 2-Layer MKL (2LMKL) and Infinite 2-Layer MKL in (Zhuang et al., 2011) (2LMKL$^{\text{INF}}$).

Eleven binary classification data sets that have been widely used in MKL literature were split evenly for training and test and were all normalized to zero mean and unit variance prior to training. 20 runs with identical settings but random weight initializations were repeated for each model. For each repetition, a new training-test split was selected randomly.

For kMLP, all results were achieved using a greedily-trained, one-hidden-layer model with the number of kernel machines ranging from 3 to 10 on the first layer for different data sets. The second layer was a single kernel machine. All kernel machines within one layer used the same Gaussian kernel, and the two kernels on the two layers differed only in kernel width $\sigma$. All hyperparameters were chosen via 5-fold cross-validation. As for the other models compared, for each data set, SVM used a Gaussian kernel. For the MKL algorithms, the base kernels contained Gaussian kernels with 10 different widths on all features and on each single feature and polynomial kernels of degree 1 to 3 on all features and on each single feature. For 2LMKL$^{\text{INF}}$, one Gaussian kernel was added to the base kernels at each iteration. Each base kernel matrix was normalized to unit trace. For $L^p$MKL, $p$ was selected from $\{2, 3, 4\}$. For MKM, the degree parameter was chosen from $\{0, 1, 2\}$. All hyperparameters were selected via 5-fold cross-validation. From Table 2, kMLP compares favorably with other models, which validates our claim that kMLP learns its own kernels nonparametrically hence can work well even without excessive kernel parameterization. Performance difference among models can be small for some data sets, which is expected since they are all rather small in size and not too challenging. Nevertheless, it is worth noting that only 2 Gaussian kernels were used for kMLP, whereas all other models except for SVM used significantly more kernels.

## APPENDIX C    PROOFS

***Proof of Lemma 4.3***. Throughout this proof we shall drop the layer index $l$ for brevity. Given that the representation satisfies Eq. 1, the idea is to first collect enough information about the returned $f^\star = (w^\star, b^\star)$ such that we can compute $\hat{R}(f^\star) + \tau\|w^\star\|_H$ and then show that for any other $F'(S)$ satisfying the condition in the lemma, suppose the learning paradigm returns $f' = (w', b') \in \mathcal{F}_{A'}$, then $\hat{R}(f') + \tau\|w'\|_H \geq \hat{R}(f^\star) + \tau\|w^\star\|_H$. We now start the formal proof.

First, note that in the optimal representation, i.e., an $F(S)$ such that Eq. 1 holds, it is easy to see that $\|\phi(F(x_-)) - \phi(F(x_+))\|_H$ is maximized over all representations for all $x_-, x_+$.

Moreover, note that given the representation is optimal, we have $\phi(F(x)) = \phi(F(x'))$ if $y = y'$ and $\phi(F(x)) \neq \phi(F(x'))$ if $y \neq y'$: Indeed, by Cauchy-Schwarz inequality, for all $x, x' \in S$, $k(F(x), F(x')) = \langle\phi(F(x)), \phi(F(x'))\rangle_H \leq \|\phi(F(x))\|_H\|\phi(F(x'))\|_H$ and the equality holds if

and only if $\phi(F(x)) = p\phi(F(x'))$ for some real constant $p$. Using the assumption on $k$, namely, that $\|\phi(F(x))\|_H = \sqrt{c}$ for all $F(x)$, we further conclude that the equality holds if and only if $p = 1$. And the second half of the claim follows simply from $c > a$. Thus, all examples from the $+$ and $-$ class can be viewed as one vector $\phi(F(x_+))$ and $\phi(F(x_-))$, respectively.

The returned hyperplane $f^\star$ cannot pass both $F(x_+)$ and $F(x_-)$, i.e., $f^\star(F(x_+)) = 0$ and $f^\star(F(x_-)) = 0$ cannot happen simultaneously since if so, first subtract $b^\star$, rotate while keeping $\|w^\star\|_H$ unchanged and add some suitable $b'$ to get a new $f'$ such that $f'(F(x_-)) < 0$ and $f'(F(x_+)) > 0$, then it is easy to see that $\hat{R}(f') + \tau\|w'\|_H < \hat{R}(f^\star) + \tau\|w^\star\|_H$. But by construction of the learning paradigm, this is not possible.

Now suppose the learning paradigm returns an $f^\star$ such that

$$
\begin{aligned}
y_+ f^\star(F(x_+)) + y_- f^\star(F(x_-)) &= \langle \phi(F(x_+)) - \phi(F(x_-)), w^\star \rangle_H \\
&= \|\phi(F(x_+)) - \phi(F(x_-))\|_H \|w^\star\|_H \cos\theta_{F,w^\star} =: \zeta.
\end{aligned}
\tag{4}
$$

First note that for an arbitrary $\theta_{F,w^\star}$, $\zeta$ is less than or equal to 2 since one can always adjust $b^\star$ such that $y_+ f^\star(F(x_+)) = y_- f^\star(F(x_-))$ without changing $\zeta$ and hence having a larger $\zeta$ will not further reduce $\hat{R}(f^\star)$, which is 0 when $\zeta = 2$, but will result in a larger $\|w^\star\|_H$ according to Eq. 4. On the other hand, $\theta_{F,w^\star}$ must be 0 since this gives the largest $\zeta$ with the smallest $\|w^\star\|_H$. Indeed, if the returned $f^\star$ does not satisfy $\theta_{F,w^\star} = 0$, one could always shift, rotate while keeping $\|w^\star\|_H$ fixed and then shift the hyperplane back to produce another $f'$ with $\theta_{F,w'} = 0$ and this $f'$ results in a larger $\zeta$ if $\zeta < 2$ or the same $\zeta$ if $\zeta = 2$ but a smaller $\|w'\|_H$ by rescaling. Hence $\hat{R}(f') + \tau\|w'\|_H < \hat{R}(f^\star) + \tau\|w^\star\|_H$ but again, this is impossible.

Together with what we have shown earlier, we conclude that $2 \geq \zeta > 0$. Then for some $t \in \mathbb{R}$, we have

$$
\hat{R}(f^\star) = \kappa \max(0, 1 - t) + (1 - \kappa)\max(0, 1 - (\zeta - t)).
$$

First note that we can choose $t$ freely while keeping $w^\star$ fixed by changing $b^\star$. If $\kappa = 1/2$, we have

$$
\hat{R}(f^\star) = \begin{cases} 1 - \zeta/2, & \text{if } 1 \geq t \geq \zeta - 1 \\ \frac{1}{2}(1 + t - \zeta), & \text{if } t > 1 \\ \frac{1}{2}(1 - t), & \text{if } t < \zeta - 1. \end{cases}
$$

Evidently, the last two cases both result in $\hat{R}(f^\star) > 1 - \zeta/2$ hence $f^\star$ must produce a $t$ in $[\zeta - 1, 1]$ and $\hat{R}(f^\star) = 1 - \zeta/2$.

Now, when $\kappa \neq 1/2$, first observe that if $1 \geq t \geq \zeta - 1$,

$$
\begin{aligned}
\hat{R}(f^\star) &= \kappa(1 - t) + (1 - \kappa)(t - (\zeta - 1)) \\
&= (1 - 2\kappa)t + \kappa - (1 - \kappa)(\zeta - 1).
\end{aligned}
$$

If $\kappa > 1/2$, $\hat{R}(f^\star)$ decreases in $t$ hence $t$ must be 1 for $f^\star$, which implies $\hat{R}(f^\star) = (1 - \kappa)(2 - \zeta)$. Similarly, if $\kappa < 1/2$, $t = \zeta - 1$ and hence $\hat{R}(f^\star) = \kappa(2 - \zeta)$.

Now suppose $t \geq 1$, $\hat{R}(f^\star) = (1 - \kappa)(1 + t - \zeta)$, which increases in $t$ and hence $t = 1$ and $\hat{R}(f^\star) = (1 - \kappa)(2 - \zeta)$. If $\kappa < 1/2$, since $(1 - \kappa)(2 - \zeta) > \kappa(2 - \zeta)$, this combination of $\kappa$ and $t$ contradicts the optimality assumption of $f^\star$.

If $t \leq \zeta - 1$, $\hat{R}(f^\star) = \kappa(1 - t) = \kappa(2 - \zeta)$, where the second equality is because $\hat{R}(f^\star)$ decreases in $t$. Again, $\kappa > 1/2$ leads to a contradiction.

Combining all cases, we have

$$
\begin{aligned}
\hat{R}(f^\star) + \tau\|w^\star\|_H &= \min(\kappa, 1 - \kappa)(2 - \zeta) + \frac{\tau\zeta}{\|\phi(F(x_+)) - \phi(F(x_-))\|_H} \\
&= \min(\kappa, 1 - \kappa)(2 - \zeta) + \frac{\tau\zeta}{\sqrt{2(c - a)}} \\
&= 2\min(\kappa, 1 - \kappa) + \left(\frac{\tau}{\sqrt{2(c - a)}} - \min(\kappa, 1 - \kappa)\right)\zeta,
\end{aligned}
$$

which, by the assumption on $\tau$, strictly decreases in $\zeta$ over $(0, 2]$. Hence the returned $f^\star$ must satisfy $\zeta = 2$, which implies $\hat{R}(f^\star) = 0$ and we have

$$\hat{R}(f^\star) + \tau\|w^\star\|_H = \frac{\sqrt{2}\tau}{\sqrt{c-a}}.$$

Now, for any other $F'(S)$, suppose the learning paradigm returns $f'$. Let $x_+^{w'}, x_-^{w'}$ be the pair of examples with the largest $f'(F'(x_+)) - f'(F'(x_-))$. We have

$$y_+^{w'} f'(F'(x_+^{w'})) + y_-^{w'} f'(F'(x_-^{w'})) = \|\phi(F'(x_+^{w'})) - \phi(F'(x_-^{w'}))\|_H \|w'\|_H \cos\theta_{F',w'} =: \zeta'.$$

Then

$$\hat{R}(f') + \tau\|w'\|_H \geq \tau\|w'\|_H$$

$$= \frac{\tau\zeta'}{\|\phi(F'(x_+^{w'})) - \phi(F'(x_-^{w'}))\|_H \cos\theta_{F',w'}}$$

$$\geq \frac{\tau|\zeta'|}{\|\phi(F'(x_+^{w'})) - \phi(F'(x_-^{w'}))\|_H}$$

$$\geq \frac{2\tau}{\sqrt{2(c-a)}}$$

$$\geq \hat{R}(f^\star) + \tau\|w^\star\|_H,$$

where we have used the assumption that there exists $x_-, x_+$ with $y_- f'(F'(x_-)), y_+ f'(F'(x_+)) \geq 1$. This proves the desired result. $\qquad\square$

**Lemma C.1.** *Suppose $f_1 \in \mathcal{F}_1, \ldots, f_d \in \mathcal{F}_d$ are elements from sets of real-valued functions defined on all of $X_1, X_2, \ldots, X_m$, where $X_j \subset \mathbb{R}^d$ for all $j$, $\mathcal{F} \subset \mathcal{F}_1 \times \cdots \times \mathcal{F}_d$. For $f \in \mathcal{F}$, define $\omega \circ f :$ $X_1 \times \cdots \times X_m \times Y \to \mathbb{R}$ as $(x_1, \ldots, x_m, y) \mapsto \omega(f_1(x_1), \ldots, f_d(x_1), f_1(x_2), \ldots, f_d(x_m), y)$, where $\omega : \mathbb{R}^{md} \times Y \to \mathbb{R}^+ \cup \{0\}$ is bounded and $L$-Lipschitz for each $y \in Y$ with respect to the Euclidean metric on $\mathbb{R}^{md}$. Let $\omega \circ \mathcal{F} = \{\omega \circ f : f \in \mathcal{F}\}$. Denote the Gaussian complexity of $\mathcal{F}_i$ on $X_j$ as $\mathcal{G}_N^j(\mathcal{F}_i)$, if the $\mathcal{F}_i$ are closed under negation, i.e., for all $i$, if $f \in \mathcal{F}_i$, then $-f \in \mathcal{F}_i$, we have*

$$\mathcal{G}_N(\omega \circ \mathcal{F}) \leq 2L \sum_{i=1}^{d} \sum_{j=1}^{m} \mathcal{G}_N^j(\mathcal{F}_i). \tag{5}$$

*In particular, for all $j$, if the $x_n^j$ upon which the Gaussian complexities of the $\mathcal{F}_i$ are evaluated are sets of i.i.d. random elements with the same distribution, we have $\mathcal{G}_N^1(\mathcal{F}_i) = \cdots = \mathcal{G}_N^m(\mathcal{F}_i) := \mathcal{G}_N(\mathcal{F}_i)$ for all $i$ and Eq. 5 becomes*

$$\mathcal{G}_N(\omega \circ \mathcal{F}) \leq 2mL \sum_{i=1}^{d} \mathcal{G}_N(\mathcal{F}_i).$$

This lemma is a generalization of a result on the Gaussian complexity of Lipschitz functions on $\mathbb{R}^k$ from (Bartlett & Mendelson, 2002). And the technique used in the following proof is also adapted from there.

*Proof.* For the sake of brevity, we prove the case where $m = 2$. The general case uses exactly the same technique except that the notations would be more cumbersome.

Let $\mathcal{F}$ be indexed by $\mathcal{A}$. Without loss of generality, assume $|\mathcal{A}| < \infty$. Define

$$X_\alpha = \sum_{n=1}^{N} \omega(f_{\alpha,1}(x_n), \ldots, f_{\alpha,d}(x_n'), y_n) g_n;$$

$$Y_\alpha = L \sum_{n=1}^{N} \sum_{i=1}^{d} (f_{\alpha,i}(x_n) g_{n,i} + f_{\alpha,i}(x_n') g_{N+n,i}),$$

where $\alpha \in \mathcal{A}$, the $(x_n, x'_n)$ are a sample of size $N$ from $X_1 \times X_2$ and $g_1, \ldots, g_N, g_{1,1}, \ldots, g_{2N,d}$ are i.i.d. standard normal random variables.

Let arbitrary $\alpha, \beta \in \mathcal{A}$ be given, define $\|X_\alpha - X_\beta\|_2^2 = \mathbf{E}(X_\alpha - X_\beta)^2$, where the expectation is taken over the $g_n$. Define $\|Y_\alpha - Y_\beta\|_2^2$ similarly and we have

$$
\begin{aligned}
\|X_\alpha - X_\beta\|_2^2 &= \sum_{n=1}^{N} \Big( \omega(f_{\alpha,1}(x_n), \ldots, f_{\alpha,d}(x'_n), y_n) - \omega(f_{\beta,1}(x_n), \ldots, f_{\beta,d}(x'_n), y_n) \Big)^2 \\
&\leq L^2 \sum_{n=1}^{N} \sum_{i=1}^{d} \Big( (f_{\alpha,i}(x_n) - f_{\beta,i}(x_n))^2 + (f_{\alpha,i}(x'_n) - f_{\beta,i}(x'_n))^2 \Big) \\
&= \|Y_\alpha - Y_\beta\|_2^2 .
\end{aligned}
$$

By Slepian's lemma (Pisier, 1999) and since the $\mathcal{F}_i$ are closed under negation,

$$
\begin{aligned}
\frac{N}{2}\hat{\mathcal{G}}_N(\omega \circ \mathcal{F}) = \mathbf{E}_{g_n} \sup_{\alpha \in \mathcal{A}} X_\alpha \\
\leq 2\mathbf{E}_{g_{n,i}, g_{N+n,i}} \sup_{\alpha \in \mathcal{A}} Y_\alpha \\
\leq \frac{N}{2} 2L \sum_{i=1}^{d} (\hat{G}_N^1(\mathcal{F}_i) + \hat{G}_N^2(\mathcal{F}_i)).
\end{aligned}
$$

Taking the expectation of the $x_n$ and $x'_n$ on both sides, we have

$$
\frac{N}{2}\mathcal{G}_N(\omega \circ \mathcal{F}) \leq \frac{N}{2} 2L \sum_{i=1}^{d} (G_N^1(\mathcal{F}_i) + G_N^2(\mathcal{F}_i)).
$$

$\square$

***Proof of Lemma 4.4***. Normalize $\ell_{l-1}$ to $[0,1]$ by dividing $2\max(|c|,|a|)$. Then the loss function becomes

$$
\ell_{l-1}\big(F^{(l-1)}, (x_m, y_m), (x_n, y_n)\big) = \frac{1}{2\max(|c|,|a|)} \Big| k^{(l)}\big(F^{(l-1)}(x_m), F^{(l-1)}(x_n)\big) - (G^\star)_{mn} \Big|.
$$

For each fixed $(G^\star)_{mn}$,

$$
\begin{aligned}
&\Big| \ell_{l-1}\big(F^{(l-1)}, (x_m, y_m), (x_n, y_n)\big) - \ell_{l-1}\big(F^{(l-1)}, (x'_m, y'_m), (x'_n, y'_n)\big) \Big| \\
&\leq \frac{1}{2\max(|c|,|a|)} \Big| k^{(l)}\big(F^{(l-1)}(x_m), F^{(l-1)}(x_n)\big) - k^{(l)}\big(F^{(l-1)}(x'_m), F^{(l-1)}(x'_n)\big) \Big| \\
&\leq \frac{1}{2\max(|c|,|a|)} \Big( \Big| k^{(l)}\big(F^{(l-1)}(x_m), F^{(l-1)}(x_n)\big) - k^{(l)}\big(F^{(l-1)}(x_m), F^{(l-1)}(x'_n)\big) \Big| \\
&\quad + \Big| k^{(l)}\big(F^{(l-1)}(x_m), F^{(l-1)}(x'_n)\big) - k^{(l)}\big(F^{(l-1)}(x'_m), F^{(l-1)}(x'_n)\big) \Big| \Big) \\
&\leq \frac{L^{(l)}}{2\max(|c|,|a|)} \Big( \big\| F^{(l-1)}(x_n) - F^{(l-1)}(x'_n) \big\|_2 + \big\| F^{(l-1)}(x_m) - F^{(l-1)}(x'_m) \big\|_2 \Big) \\
&\leq \frac{L^{(l)}}{\max(|c|,|a|)} \Big\| \big( F^{(l-1)}(x_n) - F^{(l-1)}(x'_n), F^{(l-1)}(x_m) - F^{(l-1)}(x'_m) \big) \Big\|_2 .
\end{aligned}
$$

Hence $\ell_{l-1}$ is $L^{(l)}/\max(|c|,|a|)$-Lipschitz in $(F^{(l-1)}(x_m), F^{(l-1)}(x_n))$ with respect to the Euclidean metric on $\mathbb{R}^{2d_{l-1}}$ for each $(G^\star)_{mn}$.

The result follows from Lemma C.1 with $m = 2, d = d_{l-1}$ and Corollary 15 in (Bartlett & Mendelson, 2002). $\square$

***Proof of Lemma 4.5***. This proof uses essentially the same idea as that of Lemma 4.3. Due to the complete dependence of $k^{(l)}(x, y)$ on $\|x - y\|_2$, we can rewrite $k^{(l)}\Big(F^{(l-1)}(F(x_m)), F^{(l-1)}(F(x_n))\Big)$

as $h^{(l)}\left(\left\|F^{(l-1)}(F(x_m)) - F^{(l-1)}(F(x_n))\right\|_2\right)$ for some $h^{(l)}$. Define $\mu_{mn}^{(l-1)} = \|\phi^{(l-1)}(F(x_m)) - \phi^{(l-1)}(F(x_n))\|_{H_{l-1}}$, we have

$$
\begin{aligned}
&\hat{R}_{l-1}(F^{(l-1)}) \\
&= \frac{1}{N^2} \sum_{m,n=1}^{N} \left| h^{(l)}\left(\left\|F^{(l-1)}(F(x_m)) - F^{(l-1)}(F(x_n))\right\|_2\right) - (G^\star)_{mn} \right| \\
&= \frac{1}{N^2} \sum_{m,n=1}^{N} \left| h^{(l)}\left(\sqrt{\sum_{j=1}^{d_{l-1}} \left(f_j^{(l-1)}(F(x_m)) - f_j^{(l-1)}(F(x_n))\right)^2}\right) - (G^\star)_{mn} \right| \\
&= \frac{1}{N^2} \sum_{m,n=1}^{N} \left| h^{(l)}\left(\sqrt{\sum_{j=1}^{d_{l-1}} \left(\mu_{mn}^{(l-1)} \|w_{f_j^{(l-1)}}\|_{H_{l-1}} \cos\theta_{F,w_{f_j^{(l-1)}}}\right)^2}\right) - (G^\star)_{mn} \right|.
\end{aligned}
$$

Given that the representation is optimal, we have

$$
\begin{aligned}
&\hat{R}_{l-1}(F^{(l-1)}) + \tau \max_{1 \le j \le d_{l-1}} \|w_{f_j^{(l-1)}}\|_{H_{l-1}} \\
&= \frac{1}{N^2} \sum_{y_m \ne y_n} \left( h^{(l)}\left(\sqrt{\sum_{j=1}^{d_{l-1}} \left(\mu_{mn}^{(l-1)} \|w_{f_j^{(l-1)}}\|_{H_{l-1}} \cos\theta_{F,w_{f_j^{(l-1)}}}\right)^2}\right) - a \right) \\
&\quad + \tau \max_{1 \le j \le d_{l-1}} \|w_{f_j^{(l-1)}}\|_{H_{l-1}}.
\end{aligned}
$$

The returned $F^{(l-1)\star}$ must satisfy

$$
\sqrt{\sum_{j=1}^{d_{l-1}} \left(\mu_{mn}^{(l-1)} \|w_{f_j^{(l-1)\star}}\|_{H_{l-1}} \cos\theta_{F,w_{f_j^{(l-1)\star}}}\right)^2} \le \eta;
$$
$$
\|w_{f_1^{(l-1)\star}}\|_{H_{l-1}} = \cdots = \|w_{f_{d_{l-1}}^{(l-1)\star}}\|_{H_{l-1}} =: \|w^\star\|_{H_{l-1}};
$$
$$
\cos\theta_{F,w^\star} = 1.
$$

The first observation is trivial since if otherwise, one can always reduce the largest $\|w_{f_j^{(l-1)\star}}\|_{H_{l-1}}$ to obtain equality to $\eta$, this gives the same $\hat{R}_{l-1}(F^{(l-1)\star})$ with a smaller $\tau \max_j \|w_{f_j^{(l-1)\star}}\|_{H_{l-1}}$. Note that if during shrinking the largest $\|w_{f_j^{(l-1)\star}}\|_{H_{l-1}}$, this element ceases to be the largest among all $j$, we shall continue the process with the new largest instead. To see the rest of the claim, note that for the largest of the $\|w_{f_j^{(l-1)\star}}\|_{H_{l-1}}$, we must have $\theta_{F,w_{f_j^{(l-1)\star}}} = 0$ since if not, one could shift and rotate the hyperplane and again obtain the same $\hat{R}_{l-1}(F^{(l-1)\star})$ with a smaller $\tau \max_j \|w_{f_j^{(l-1)\star}}\|_{H_{l-1}}$. Reducing the largest $\|w_{f_j^{(l-1)\star}}\|_{H_{l-1}}$ and increasing the second largest by scaling, one would get a smaller risk. It is immediate that the minimal risk (w.r.t. only the first and second largest $\|w_{f_j^{(l-1)\star}}\|_{H_{l-1}}$) is attained when the largest and the second largest $\|w_{f_j^{(l-1)\star}}\|_{H_{l-1}}$ are equal. Then a similar argument as before gives $\theta_{F,w_{f_j^{(l-1)\star}}} = 0$ for both of them. The rest of the claim follows via repeatedly applying this argument to all the $\|w_{f_j^{(l-1)\star}}\|_{H_{l-1}}$.

Define

$$
|f^\star(F(x_+)) - f^\star(F(x_-))| = \|\phi^{(l-1)}(F(x_+)) - \phi^{(l-1)}(F(x_-))\|_{H_{l-1}} \|w^\star\|_{H_{l-1}}.
$$

Then we have

$$\hat{R}_{l-1}(F^{(l-1)\star}) + \tau \max_{1 \leq j \leq d_{l-1}} \|w_{f_j^{(l-1)\star}}\|_{H_{l-1}}$$

$$= \psi \left( h^{(l)} \left( \sqrt{d_{l-1}} |f^\star(F(x_+)) - f^\star(F(x_-))| \right) - a \right)$$

$$+ \frac{\tau |f^\star(F(x_+)) - f^\star(F(x_-))|}{\|\phi^{(l-1)}(F(x_+)) - \phi^{(l-1)}(F(x_-))\|_{H_{l-1}}}$$

$$= \psi \left( h^{(l)} \left( \sqrt{d_{l-1}} |f^\star(F(x_+)) - f^\star(F(x_-))| \right) - a \right)$$

$$+ \frac{\tau |f^\star(F(x_+)) - f^\star(F(x_-))|}{\sqrt{2(c-a)}}.$$

As we have shown, $\sqrt{d_{l-1}} |f^\star(F(x_+)) - f^\star(F(x_-))| \in [0, \eta]$. Let $\lambda = |f^\star(F(x_+)) - f^\star(F(x_-))|$ and differentiate the r.h.s. of the above equation w.r.t. $\lambda$ and using the assumption on $\tau$, we have

$$\sqrt{d_{l-1}} \psi \frac{dh^{(l)}(t)}{dt} + \frac{\tau}{\sqrt{2(c-a)}} < 0.$$

Hence the overall risk decreases in $\lambda$ over $[0, \eta/\sqrt{d_{l-1}}]$, which implies that the returned $F^{(l-1)\star}$ must have $\lambda = \eta/\sqrt{d_{l-1}}$ and

$$\hat{R}_{l-1}(F^{(l-1)\star}) + \tau \max_{1 \leq j \leq d_{l-1}} \|w_{f_j^{(l-1)\star}}\|_{H_{l-1}} = \frac{\tau \eta}{\sqrt{2d_{l-1}(c-a)}}.$$

Now for any other representation $S'_{l-2} = F'(S)$, suppose the learning paradigm returns $F^{(l-1)'}$. Assume without loss of generality that the largest $f_j^{(l-1)'}(F'(x_+)) - f_j^{(l-1)'}(F'(x_-))$ over all $j$, $x_+$ and $x_-$ is attained at $j = 1$ and write $w' = w_{f_1^{(l-1)'}}$, $f' = f_1^{(l-1)'}$ for convenience. Let $x_+^{w'}$, $x_-^{w'}$ be the pair with the largest $f'(F'(x_+)) - f'(F'(x_-))$. Note that the assumption on $F'$ implies that $f'(F'(x_+)) - f'(F'(x_-)) \geq \eta/\sqrt{d_{l-1}}$. Then we have

$$\hat{R}_{l-1}(F^{(l-1)'}) + \tau \max_{1 \leq j \leq d_{l-1}} \|w_{f_j^{(l-1)'}}\|_{H_{l-1}}$$

$$\geq \tau \|w'\|_{H_{l-1}}$$

$$= \frac{\tau |f'(F'(x_+)) - f'(F'(x_-))|}{\|\phi^{(l-1)}(F'(x_+)) - \phi^{(l-1)}(F'(x_-))\|_{H_{l-1}} |\cos \theta_{F', w'}|}$$

$$\geq \frac{\tau \eta}{\sqrt{2d_{l-1}(c-a)}}$$

$$= \hat{R}_{l-1}(F^{(l-1)\star}) + \tau \max_{1 \leq j \leq d_{l-1}} \|w_{f_j^{(l-1)\star}}\|_{H_{l-1}},$$

proving the lemma. □

***Proof of Lemma 4.6.*** First, it is trivial that the so-defined $s$ metric is indeed a metric. In particular, it satisfies the triangle inequality. For $i = 2, \ldots, l$,

$$\left\| F^{(i)*} \circ F^{(i-1)*} - T_i \circ T_{i-1} \right\|_s$$

$$\leq \left\| F^{(i)*} \circ F^{(i-1)*} - F^{(i)*} \circ T_{i-1} \right\|_s + \left\| F^{(i)*} \circ T_{i-1} - T_i \circ T_{i-1} \right\|_s$$

$$\leq \sup_{x \in X_i} \sqrt{\sum_{j=1}^{d_i} \left( f_j^{(i)*} \circ F^{(i-1)*}(x) - f_j^{(i)*} \circ T_{i-1}(x) \right)^2 + \epsilon_i}$$

$$= \sup_{x \in X_i} \sqrt{\sum_{j=1}^{d_i} \left( \left\langle f_j^{(i)*}(\lambda), k^{(i)}(\lambda, F^{(i-1)*}(x)) - k^{(i)}(\lambda, T_{i-1}(x)) \right\rangle_{H_i} \right)^2 + \epsilon_i}$$

$$\leq \sup_{x \in X_i} \sqrt{\sum_{j=1}^{d_i} \left( \left\| f_j^{(i)*} \right\|_{H_i} \left\| k^{(i)}(\lambda, F^{(i-1)*}(x)) - k^{(i)}(\lambda, T_{i-1}(x)) \right\|_{H_i} \right)^2 + \epsilon_i}$$

$$= \sup_{x \in X_i} \left\| k^{(i)}(\lambda, F^{(i-1)*}(x)) - k^{(i)}(\lambda, T_{i-1}(x)) \right\|_{H_i} \sqrt{\sum_{j=1}^{d_i} \left\| f_j^{(i)*} \right\|_{H_i}^2 + \epsilon_i}$$

$$\leq \sup_{x \in X_i} \sqrt{\left( L_{F^{(i-1)*}(x)}^{(i)} + L_{T_{i-1}(x)}^{(i)} \right) \left\| F^{(i-1)*}(x) - T_{i-1}(x) \right\|_2} \sqrt{\sum_{j=1}^{d_i} \left\| f_j^{(i)*} \right\|_{H_i}^2 + \epsilon_i}$$

$$\leq \sqrt{2 L^{(i)} \sum_{j=1}^{d_i} \left\| f_j^{(i)*} \right\|_{H_i}^2} \sqrt{\left\| F^{(i-1)*} - T_{i-1} \right\|_s} + \epsilon_i,$$

which proves the lemma. $\qquad\square$

***Proof of Lemma B.1.*** Since $\Omega$ and $\mathcal{F}_2$ are both closed under negation, we have

$$\hat{\mathcal{G}}_N(\mathcal{F}_2 \circ \mathcal{F}_1) = \mathbf{E} \sup_{\boldsymbol{\alpha}, F} \left| \frac{2}{N} \sum_{n=1}^{N} \sum_{\nu=1}^{m} \alpha_\nu k(F(x_\nu), F(x_n)) g_n \right|$$

$$\leq \mathbf{E} \sup_{\boldsymbol{\alpha}, F} \frac{2}{N} \sum_{\nu=1}^{m} |\alpha_\nu| \sum_{n=1}^{N} k(F(x_\nu), F(x_n)) g_n$$

$$\leq \frac{2}{N} A \mathbf{E} \sup_{F} \max_{\nu} \sum_{n=1}^{N} k(F(x_\nu), F(x_n)) g_n$$

$$= \frac{2}{N} A \mathbf{E} \sup_{F} \max_{\nu} \sum_{n=1}^{N} \left( k(F(x_\nu), F(x_n)) - k(F(x_\nu), 0) \right) g_n$$

$$\quad + \frac{2}{N} A \mathbf{E} \sup_{F} \max_{\nu} \sum_{n=1}^{N} k(F(x_\nu), 0) g_n$$

$$\leq \frac{2}{N} A \mathbf{E} \sup_{F} \max_{\nu} \sum_{n=1}^{N} L_{F(x_\nu)} \|F(x_n)\|_2 g_n$$

$$\leq \frac{2}{N} A L \mathbf{E} \sup_{F} \sum_{n=1}^{N} \|F(x_n)\|_2 g_n$$

$$\leq \frac{2}{N} A L \mathbf{E} \sup_{F} \sum_{n=1}^{N} \|F(x_n)\|_1 g_n$$

$$\leq A L d_1 \hat{\mathcal{G}}_N(\Omega).$$

Taking expectation with respect to the $x_n$ finishes the proof. $\qquad\square$

***Proof of Lemma B.3.*** For $i \geq 1$, given the assumption that $d_{i-1} = d_i$, we first show that one can find an initialization for layer $i$, denoted $F^{(i)\circ}$, such that $F^{(i)\circ}(S) = F^{(i-1)}(S)$.

Without loss of generality, assume $i = 1$, then it amounts to showing that a set of parameters can be found for the kernel machines $f_1^{(1)}, \ldots, f_{d_1}^{(1)}$ of $F^{(1)}$ such that when these kernel machines are initialized with this certain set of parameters, this layer agrees with the identity function when restricted to $S$. Recall that $f_j^{(i)}(x) = \sum_{n=1}^{N} \alpha_{jn}^{(i)} k^{(i)}(x_n, x)$ and $\{\alpha_{jn}^{(i)} \in \mathbb{R} \mid n = 1, 2, \ldots, N\}$ is the set of trainable parameters for this kernel machine.

Solve equation $AG = D$ for $A$, where $A$ is the $d_0 \times N$ matrix of parameters defined as $(A)_{nm} = \alpha_{nm}^{(1)}$, $G$ is the $N \times N$ kernel matrix defined as $(G)_{nm} = k^{(1)}(x_n, x_m)$, $D$ is a $d_0 \times N$ matrix whose $n$th column is the $n$th example $x_n = (x_{n1}, x_{n2}, \ldots, x_{nd_0})^\top$ for $n = 1, 2, \ldots, N$. It is straightforward that the set of parameters $A$ initializes $F^{(1)}$ to the desired $F^{(1)\circ}$. Moreover, $A$ can always be solved in closed-form by inverting $G$ since $k^{(1)}$ being PD guarantees that $G$ is invertible.

Combining the above result and the defining property of gradient of the loss function with respect to the parameters, i.e., that gradient points at the direction of the greatest rate of increase of the loss function, we then have

$$\ell(F^{(0)}) = \ell(F^{(1)\circ}) \geq \ell(F^{(1)*}).$$

$\qquad\square$

***Proof of Lemma B.4.*** In this proof we shall again drop the layer index $l$ for brevity.

Identify $f \in \mathcal{F}_A$ using $w, b$ upto $\mathbb{R}$-scalar multiplication. Fix an $F$ such that $\phi(F(S))$ is linearly separable (otherwise the learning paradigm is not applicable).

Let $x_+, x_-$ be the closest pair of examples from distinct classes in $H$. For a given $f = (w, b)$, let $x_-^w, x_+^w$ be a pair of examples from different classes that are closest to $f$ in $H$ in the sense that $f(F(x_+^w)), f(F(x_-^w))$ are the smallest in absolute value among all $x_+, x_-$, respectively. Denote the canonical form of $f$ with respect to $\phi(F(S))$ with equality attained by at least one of $x_+^w, x_-^w$ as $f_F = (w_F, b_F)$, i.e., $f_F$ satisfies $y f_F(x) \geq 1$ for all examples in $(x_n, y_n)_{n=1}^N$ and for at least one of $x_-^w, x_+^w$, the equality is attained. Define $A_{f_F} := \inf_{b_F \in \mathbb{R}} \|w_F\|_H$.

Suppose the algorithm returns $f^\star = (w^\star, b^\star)$ and recall that $A$ is the smallest such that $f^\star \in \mathcal{F}_A$, then for all $f = (w, b) \in \mathcal{F}_A$ such that $\hat{R}(f) = 0$,

$$A \geq \|w\|_H \geq \|w_F\|_H \geq A_{f_F}.$$

And it is easy to see that the last equality holds if and only if $b_F$ is such that $y_-^w f_F(x_-^w) = y_+^w f_F(x_+^w)$. Then $f_F \in \mathcal{F}_A$. Since $\hat{R}(f_F) = 0$ and we have the freedom to choose $b_F$ such that the last equality holds, by construction of the learning paradigm and the minimality of $A$, we have

$$A = \min_{\substack{f \in \mathcal{F}_A \\ \hat{R}(f) = 0}} A_{f_F} = A_{f_F^\star}.$$

In canonical form, for any $f_F \in \mathcal{F}_A$, choose $b_F$ such that $\|w_F\|_H = A_{f_F}$, then

$$\frac{2}{A_{f_F}} = \frac{1}{A_{f_F}} \langle \phi(F(x_+^w)) - \phi(F(x_-^w)), w_F \rangle_H$$

$$\leq \frac{1}{A_{f_F}} \langle \phi(F(x_+)) - \phi(F(x_-)), w_F \rangle_H = \big\|\phi(F(x_+)) - \phi(F(x_-))\big\|_H \cos \theta_{F,w}.$$

By construction of the learning paradigm, $f^\star$ must make the equality hold and satisfy $\theta_{F,w^\star} = 0$. Hence we have

$$A = A_{f_F^\star} = \frac{2}{\big\|\phi(F(x_+)) - \phi(F(x_-))\big\|_H},$$

which proves that, as a function of $F$, $A$ achieves its minimum if and only if $F$ maximizes $\left\|\phi(F(x_+)) - \phi(F(x_-))\right\|_H$. Since

$$\arg\max_F \left\|\phi(F(x_+)) - \phi(F(x_-))\right\|_H$$

$$= \arg\max_F \left(\left\|\phi(F(x_+))\right\|_H^2 + \left\|\phi(F(x_-))\right\|_H^2 - 2\langle\phi(F(x_+)), \phi(F(x_-))\rangle_H\right)$$

$$= \arg\min_F k\big(F(x_+), F(x_-)\big),$$

where we have used the assumption on $k$, namely, that $k(x, x) = \langle\phi(x), \phi(x)\rangle_H = \|\phi(x)\|_H^2 = c$, for all $x$. It immediately follows that any minimizer $F$ of $A$ must minimize $k(F(x_+), F(x_-))$ for all pairs of examples from opposite classes. This proves the desired result. $\qquad\square$

