# OpenReview forum: "Learning Backpropagation-Free Deep Architectures with Kernels"
_ICLR.cc/2019/Conference_

### Official Review · AnonReviewer3 · 2018-11-02
**interesting theoretical analysis of layer-wise training of kernel-based neural networks, concerns about practicality**

**Rating:** 5
**Confidence:** 3

**Review:**

Summary: The paper considers so-called kernel neural networks where the non-linear activation function at each neuron is replaced by a kernelized linear operation, and analyses a layer-wise training scheme to train such networks. The theoretical claims are that (i) the optimal representation at each hidden layer can be determined by getting the similarity between two kernel matrices and (ii) this procedure gives a more interpretable training procedure and can avoid the vanishing gradient problems. Some small-scale experiments are provided.

Evaluation: I have a mixed feeling about this paper: the theoretical contributions seem interesting but its interpretation and practicality are somewhat non-intuitive and philosophically troubling, in my opinion. I did not check the proofs in the appendix so I might have missed some critical info or have not fully understood the experimental set-up.

- interpretability: it's not clear to me if this training scheme is any more interpretable than backprop training (not to mention it's not clear to me how to define interpretability for neural networks). Whether BP or any layer-wise training schemes is used, isn't the goal is to get S_{l-1} to the state where S_{l-1}s for examples of different classes are far away from each other as this is easier for the classifier?
- function representation: in section 2, fj^i(x) is parameterized as a sum of kernel values evaluated at x and the training points. It's unclear to me what is x here -- input to the network or output of the previous layer? This also has a sum over all training points, so is training kMLPs in a layer-wise fashion more efficient than traditional kernel methods?
- training scheme: what is the order of layers being trained? input to output or output to input? I'm slightly hazy on how to obtain F^{(l-1)}(S) to compute G_{l-1}.
- the intuition of layer-wise optimality: on page 4, the paper states that "the global min of R_l wrt S_{l-1} can be explicitly identified prior to any training" but intuitively this must condition on some known function/function class F^(l). Could you please enlighten me on this?
- the experiments are of small-scale and, as the paper pointed out, only demonstrating the concepts. What are the main practical difficulties preventing this from being applied to bigger networks/bigger datasets?
- vanishing gradients: I'm not clear how layer-wise training can avoid this issue - could you please explain this?
- some typos: p1 emplying -> employing, p4 supress -> suppress, p5 represnetation -> representation

---

> ### Author Response · Authors · 2018-11-08
> **Reply to Reviewer 3 [1/2]**
>
>
> We would like to first thank Reviewer 3 for the helpful comments and questions. Our detailed reply is provided below. And we hope that it answers the questions from Reviewer 3.
>
> Comments from the reviewer are listed first with each preceded by a dash. Our replies are put in brackets.
>
> ---------------------------------------------
>
> - the theoretical contributions seem interesting but its interpretation and practicality are somewhat non-intuitive and philosophically troubling
>
> [In terms of the interpretation of the theoretical results, our work showed that, thanks to the use of kernel functions as nonlinearities in a connectionist architecture, one can concretely answer the question: What is the best representation for each hidden layer?
>
> This is the fundamental question behind training deep connectionist models [1]. Before, the most widely-accepted answer to this problem was an indirect one: Use BP. This is indirect since, despite that BP does the job of training the model well, it does not provide any interpretable or generalizable knowledge as to what defines a good hidden representation. In contrast, our Lemma 4.3 and Lemma 4.5 provided explicit and general conditions characterizing such optimal hidden representations.
>
> In terms of practicality, these theoretical results removed the need for BP and directly made possible a layer-wise learning algorithm with optimality guarantee equivalent to that offered by BP. Among works that try to replace BP, to the best of our knowledge, ours is the first to provide such an optimality guarantee, thanks to the theoretical results in this paper.]
>
> ---------------------------------------------
>
> - interpretability: it's not clear to me if this training scheme is any more interpretable than backprop training (not to mention it's not clear to me how to define interpretability for neural networks).
>
> [We are thankful that the reviewer brought up this important issue. Please see our reply to all reviewers for our response.]
>
> - Whether BP or any layer-wise training schemes is used, isn't the goal is to get S_{l-1} to the state where S_{l-1}s for examples of different classes are far away from each other as this is easier for the classifier?
>
> [For the hidden layers in NN, it is difficult to even talk about the notion of examples being "far away from each other" since there is no natural metric space in which the training can be geometrically interpreted or discussed. Of course, one may use the Euclidean space in which the hidden activation vectors live, but it is not entirely trivial (at least to us) how to prove that what backpropagation does is to push examples from distinct classes as far apart as possible in the metric of that Euclidean space.
>
> For KN trained with the proposed layer-wise algorithm, on the other hand, such an interpretation can be readily applied, as we have shown in Sections 4.2.2 and 4.2.3. And this makes its learning dynamics more transparent and straightforward than NN.]
>
> ---------------------------------------------
>
> - function representation: in section 2, f_j^i(x) is parameterized as a sum of kernel values evaluated at x and the training points. It's unclear to me what is x here -- input to the network or output of the previous layer?
>
> [Edit (Nov. 26): We have updated this section to clarify. Please refer to the newest manuscript for details.]
>
> - This also has a sum over all training points, so is training kMLPs in a layer-wise fashion more efficient than traditional kernel methods?
>
> [In terms of computational complexity, a kMLP is more demanding than a traditional kernel machine since the latter corresponds to a single node in the former. Section 4.4 provided a natural accelerating approach to mitigate this issue in practice.
>
> Nevertheless, our results in Appendix B.5.1 suggest that kMLP performs much better than traditional kernel machine and kernel machine enhanced by state-of-the-art multiple kernel learning algorithms.]
>
> ---------------------------------------------
>
> - training scheme: what is the order of layers being trained? input to output or output to input? I'm slightly hazy on how to obtain F^(l-1)(S) to compute G_{l-1}.
>
> [The training proceeds from input to output. Each layer is trained and frozen afterward. For example, one first train F^(1) to minimize some dissimilarity measure between the ideal kernel matrix G^\star and the actual kernel matrix G_1 defined as (G_1)_{mn} = k^(2)(F^(1)(x_m), F^(1)(x_n)). After the training of F^(1), freeze it and call the frozen state F^{(1)*}. Now start training F^(2) to minimize some dissimilarity measure between G^\star and kernel matrix G_2 defined as (G_2)_{mn} = k^(3)(F^(2) \circ F^{(1)*}(x_m), F^(2) \circ F^{(1)*}(x_n)). And so on.]
>
> ---------------------------------------------

---

> ### Author Response · Authors · 2018-11-08
> **Reply to Reviewer 3 [2/2]**
>
>
> - the intuition of layer-wise optimality: on page 4, the paper states that "the global min of R_l wrt S_{l-1} can be explicitly identified prior to any training" but intuitively this must condition on some known function/function class F^(l). Could you please enlighten me on this?
>
> [This is a great question and the reviewer is indeed correct. This "conditioning" takes the form of the assumption in Lemma 4.3 (and Lemma 4.5) that the actual F^(l) considered is the one returned by a learning paradigm minimizing the loss. Putting this in a more concise form, the optimal representation S^\star_{l-1} is defined as S^\star_{l-1} := argmin_{S_{l-1}} min_{F^(l)} R_l(F^(l), S_{l-1}), where the minimum over F^(l) is understood as the minimum for each fixed S_{l-1}.]
>
> ---------------------------------------------
>
> - the experiments are of small-scale and, as the paper pointed out, only demonstrating the concepts. What are the main practical difficulties preventing this from being applied to bigger networks/bigger datasets?
>
> [Edit (Nov. 26): We have added new results on the standard MNIST and Fashion-MNIST datasets. Please refer to our summary of all changes made to the manuscript for details.]
>
> ---------------------------------------------
>
> - vanishing gradients: I'm not clear how layer-wise training can avoid this issue - could you please explain this?
>
> [As the reviewer knows, vanishing gradient occurs when one has to compute gradients of the form df_l \circ ...\circ f_1(w)/dw, where f_l, ..., f_1 are layers of a deep architecture containing some nonlinear functions. Typically, the magnitude of the resulting gradient at each layer can be small due to some properties of the nonlinearities such as their ranges being between +-1 and having plateaus everywhere but in a small region around the origin, etc. And the larger the l, the smaller the magnitude of the gradient at a layer closer to the input since it would be a product of the gradients from upper layers, all of which being some numbers with small absolute value (likely less than 1).
>
> Although layer-wise training does not eliminate the computation for gradients or change these troublesome properties of the nonlinearities mentioned above, it only requires computing gradients of the form df_i(w)/dw. And since the composition of layers does not occur, the issue mentioned above would be mitigated to some extent.]
>
> ---------------------------------------------
>
> - some typos: p1 emplying -> employing, p4 supress -> suppress, p5 represnetation -> representation
>
> [We have corrected the typos in the newest manuscript. And we are truly thankful that Reviewer 3 brought them into our attention.]
>
>
>
> [1] Rumelhart, D. E., Hinton, G. E., & Williams, R. J. (1986). Learning representations by back-propagating errors. nature, 323(6088), 533.

---

> ### Author Response · Authors · 2018-11-12
> **New manuscript has been uploaded**
>
>
> This post contained a description of updates in a past version of the paper. For clarity, all the changes we have made during the rebuttal period to the manuscript are now summarized in the newest reply to all reviewers. Please refer to that for details.

---

> ### Author Response · Authors · 2018-11-15
> **New manuscript with more experimental results has been uploaded**
>
>
> This post contained a description of updates in a past version of the paper. For clarity, all the changes we have made during the rebuttal period to the manuscript are now summarized in the newest reply to all reviewers. Please refer to that for details.

---

> ### Author Response · Authors · 2018-11-19
> **New manuscript has been uploaded**
>
>
> This post contained a description of updates in a past version of the paper. For clarity, all the changes we have made during the rebuttal period to the manuscript are now summarized in the newest reply to all reviewers. Please refer to that for details.

---

> ### Author Response · Authors · 2018-11-26
> **New results on Fashion-MNIST**
>
>
> This post contained a description of updates in a past version of the paper. For clarity, all the changes we have made during the rebuttal period to the manuscript are now summarized in the newest reply to all reviewers. Please refer to that for details.

---

> ### Author Response · Authors · 2018-12-19
> **Your thoughts on our response?**
>
>
> Dear Reviewer 3,
>
> Hello! We really hope we have fully addressed your concerns in our earlier reply and it would be great if you could give us some feedback. Do you think we have fully addressed your concerns? In particular, if you think our newly-added experimental results could validate the practicality of our algorithm, could you please reconsider your rating? Thank you very much!
>
> Best regards,
> Paper 1364 authors

---

### Official Review · AnonReviewer1 · 2018-11-02
**Interesting theoretical work which could be improved by further experimental work**

**Rating:** 6
**Confidence:** 4

**Review:**

****Reply to authors' rebuttal****

Dear Authors,

Thank you very much for all the effort you have put into the rebuttal. Based on the improved theoretical and experimental results, I have decided to increase my score from 5 to 6.

Best wishes,
Rev 1


****Original review****


This paper explores integration of kernel machines with neural networks based on replacing the non-linear function represented by each neuron with a function living in some pre-defined RKHS. From the theoretical standpoint, this work is a clear improvement upon the work of Zhang et al. (2017). Authors further propose a layer-wise training algorithm based on optimisation of a particular similarity measure between embeddings based on their class assignments at each layer, which eliminates necessity of gradient-based training. However, the experimental performance of the proposed algorithm is somewhat lacking in comparison, perhaps because the authors focus on kernelised equivalents of MLPs instead of CNNs as Zhang et al.

My rating of the paper is mainly due to the lack of experimental evidence for usefulness of the layer-wise training, and absence of experimental comparison with several baselines (see details below). It is also unclear whether the structure of KNs is significantly better than that of NNs in terms of interpretability. Apart from the comments below, I would like to ask the authors to discuss relation to the following related papers:

	1) Kulkarni & Karande, 2017: "Layer-wise training of deep networks using kernel similarity" https://arxiv.org/pdf/1703.07115.pdf

	2) Scardapanea et al., 2017: "Kafnets: kernel-based non-parametric activation functions for neural networks" https://arxiv.org/pdf/1707.04035.pdf


Detailed comments:

Theory

- (Sec 4.1) Backpropagation (BP) is being criticised: BP is only a particular implementation of gradient calculation. It seems to me that your criticisms are thus more related to use of iterative gradient-based optimisation algorithms, rather than to obtaining gradients through BP?! Regarding the criticism that BP forces intermediate layers to correct for "mistakes" made by layers higher up: it seems your layer-wise algorithm attempts to learn the best possible representation in first layer, and then progresses to the next layer where it tries to correct for the potential error of the first layer and so on. In other words it seems that the errors of layers are propagated from first to last, instead of last to first as in BP, but are still being propagated in a sense. I do not immediately see why propagation forward should be preferable. Can you please further explain this point?

- It is proven in the appendix (Lemma B.3) that under certain conditions stacking additional layers never leads to degradation of training loss. Can you please clarify whether additional layers can be helpful even in the case where previous layers already succeeded in learning the optimal representation?

- (Sec 4.1) Layer-wise vs. network-wise optimality: I find the claim that BP-based learner is not aware of the network-wise optimality confusing. BP explicitly optimises for network-wise optimality and the relative contribution to the network-wise error of each weight is propagated accordingly. I suppose my confusion stems from lack of a clear description of what defines a learner "aware" or "blind" to network-wise optimality. In general, I am not convinced layer-wise optimality is a useful criterion when what we want to achieve is network-wise optimality. As you show in the appendix, if layer-wise optimality is achieved then it implies network-wise optimality; however, layer-wise optimality is only a sufficient condition and likely not a necessary one (except for the simplified scenario studied in B.3). It is thus not clear to me why layer-wise training would always be preferable to network-wise training (e.g. using BP) especially because its greedy nature might intuitively prevent learning of hierarchical representations which are commonly claimed to be key to the success of neural networks. Can you please clarify?

- (Sec 4.2) I think it would be beneficial to state in the introduction that the "risk" is with respect to the hinge loss which is common in the SVM/kernel literature but much less in the deep learning literature and thus could surprise a few people when they reach this point.
Futher questions:
	- From Lemma 4.3, it seems that the derived representation is only optimal with respect to the **upper bound** on the empirical risk (which for \tau >= 2 will be an upper bound on the population risk). I got slightly confused at this point as my interpretation of the previous text was that the representation is optimal with respect to the population risk itself. Does the upper bound have the same set of optima? Please clarify.

	- (p.5) There are two assumptions that I find somewhat restrictive. Just before Lemma 4.3 you assume that the number of points in each class must be the same. Can you comment on whether you expect the same representation to be optimal for classification problems with significantly imbalanced number of samples per class? The second assumption is after Lemma 4.4 where you state that the stationary kernel k^{l-1} should attain its infinum for all x, y s.t. || x - y || greater than some threshold. This does not hold for many of the popular kernels like RBF, Matern, or inverse multiquadric. Do you think this assumption can be relaxed?

	- (p.5) Choice of the dissimilarity measure for G: Can you provide more intuition about why you selected L^1 distance and whether you would expect different results with L^2 or other common metrics?

- (Sec 4.3) Can you please provide more detaild about the relation of the proposed objective (\hat(R)(F) + \tau max_j ||f_j||_H) to Lemmas 4.3 and 4.5 where the optimal representation was derived for functions that optimise an upper bound in terms of Gaussian complexity (e.g. is the representation that minimises risk w.r.t. the Gaussian bound also optimal with respect to functions that optimise this objective)?


Experiments

- I would appreciate addition of some standard baselines, like MLP combined with dropout or batch normalisation, and optimised with RMSProp (or similar). These would greatly help with assessing competitiveness with current SOTA results.

- It would be nice to see the relative contribution of the two main components of the paper. Specifically, an experiment which would evaluate empirical performance of KNs optimised by some form of gradient descent vs. by your layer-wise training rule would be very insightful.


Other

- (p.2, 1st par in Sec 2) [minor] You state "a kernel machine is a universal function approximator". I suppose that might be true for a certain class of kernels but not in general?! Please clarify.

- (p.2, 3rd par in Sec 2) [minor] Are you using a particular version of the representer theorem in the representation of f_j^{(i)} as linear combination of feature maps? Please clarify.

- (p.2, end of 1st par in Sec 3) L^{(i)} is defined as sup over X_i. It is not clear to me that this constant is necessarily finite and I suspect it will not be in general (it will for the RBF kernel (and most stationary kernels) used in experiments though). Finiteness of L^{(i)} is necessary for the bound in Eq. (2) to be non-vacuous. Please clarify.

- (p.3, after 1st display in Sec 4.2.1) [minor] Missing dot after "that we wish to minimise". Next sentence states "**the** optimal F" (emphasis mine) -- I am sorry if I overlooked it, but I did not notice a proof that a solution exists and is unique, and am not familiar enough with the literature to immediately see the answer. Perhaps a footnote clarifying the statement would help.

- (p.4, 1st par in Sec 4) You say "A generalisation to regression is reserved for future work". I did not expect that based on the first few pages. On high-level, it seems that generalisation to regression need not be trivial as, for example, the optimal representation derived in Lemma 4.3 and Lemma 4.5 explicitly relies on the classification nature of the problem. Can you comment on expected difficulty of extension to regression? Possibly state in the introduction that only classification is considered in this paper.
	- (p.7, 1st par in Sec 6) [related] "However they did not extend the idea to any **arbitrary** NN" (emphasis mine). Can you please be more specific here?

- (p.5-6) [minor] Last sentence in Lemmas 4.3 and 4.5 is slightly confusing. Can you rephrase please?

- (p.6) [minor] You say "the learned decision boundary would generalise better to unseen data". Can you please clarify the last sentence (e.g. being more precise about the meaning of the word "simple" in the same sentence) and provide reference for why this is necessarily the case?

---

> ### Author Response · Authors · 2018-11-08
> **Reply to Reviewer 1 [1/6]**
>
>
> First, we thank Reviewer 1 for the very insightful comments. We can see that Reviewer 1 has read through our paper thoroughly and we are truly thankful for that. We will try our best to address the concerns and answer the questions from the reviewer and we hope that the reviewer finds our reply satisfying.
>
> Comments from the reviewer are listed first with each preceded by a dash. Our replies are put in brackets.
>
> -------------------------------
> GENERAL COMMENTS:
> -------------------------------
>
> - ...which eliminates necessity of gradient-based training.
>
> [Just to clarify, our layer-wise learning algorithm only eliminates the need of obtaining gradients using BP. It is still a gradient-based optimization per se.]
>
> ---------------------------------------------
>
> - My rating of the paper is mainly due to the lack of experimental evidence for usefulness of the layer-wise training, and absence of experimental comparison with several baselines (see details below).
>
> [The objective of the current paper was to provide a comprehensive solution to the theoretical problem and therefore, majority of the time was spent on trying to achieve this goal. Nevertheless, we completely agree with the reviewer that more empirical results would complement the theory and henceforth, we are working hard to produce more results as suggested. We shall notify all reviewers once we have new results and have updated our manuscript accordingly.]
>
> ---------------------------------------------
>
> - It is also unclear whether the structure of KNs is significantly better than that of NNs in terms of interpretability.
>
> [We are thankful that the reviewer brought up this important issue. Please see our reply to all reviewers for our response.]
>
> --------------------------------
> TWO RELATED PAPERS:
> --------------------------------
>
> We were not aware of these two papers and we thank Reviewer 1 for bringing them into our attention. Below are our comments on these two works, which we have added to the newest manuscript as well.
>
> 1) Kulkarni & Karande, 2017: "Layer-wise training of deep networks using kernel similarity" https://arxiv.org/pdf/1703.07115.pdf
>
> [This work used the idea of an ideal kernel matrix to train NNs layer-by-layer. The activation of each layer, together with a kernel function that is separate from the architecture, is used to compute a kernel matrix. And the training of each layer amounts to aligning that kernel matrix to an ideal one. The ideal kernel matrix used therein is a special case of the ideal kernel matrix characterized by our Lemma 4.3 and Lemma 4.5. However, this work did not discuss or prove the optimality of the underlying hidden representations for NNs.]
>
> ---------------------------------------------
>
> 2) Scardapanea et al., 2017: "Kafnets: kernel-based non-parametric activation functions for neural networks" https://arxiv.org/pdf/1707.04035.pdf
>
> [This work explored the possibility of substituting the nonlinearities of NNs with kernel expansions. While the resulting networks are similar to our KNs, the authors did not further study specially-tailored training methods as we did in our work. Instead, the resulting models are simply optimized with gradient-based optimization together with BP.]
>
> --------------------------------
> DETAILED COMMENTS:
> --------------------------------
>
> - (Sec 4.1) Backpropagation (BP) is being criticised: BP is only a particular implementation of gradient calculation. It seems to me that your criticisms are thus more related to use of iterative gradient-based optimisation algorithms, rather than to obtaining gradients through BP?!
>
> [Our criticism for BP in Section 1 is on the scheme of obtaining gradients via BP instead of gradient-based optimization algorithms. Our layer-wise learning algorithm is also gradient-based, as pointed out earlier in our reply. To further clarify, we now provide more details backing up our comments on BP in Section 1:
>
>     1) BP can be computationally intensive and memory inefficient. This is because in standard BP, gradients for all layers have to be computed at each update. And one can either save these gradients while updating each layer (memory inefficient), or compute gradient for each layer on the fly (requires a lot of redundant computations since one has to differentiate through all layers between the output and the layer being updated). Clearly, a layer-wise learning approach mitigates this issue.
>
>     2) Obtaining gradients through BP can cause the vanishing gradient problem when the model contains a composition of many nonlinear layers. And it is clear that a layer-wise, gradient-based optimization approach is less subject to this issue since one no longer needs to differentiate through multiple layers for each gradient computation.

---

> ### Author Response · Authors · 2018-11-08
> **Reply to Reviewer 1 [2/6]**
>
>
>    3) BP and in fact, end-to-end training algorithms in general, turns the model into a black box in the following senses. First, they make the design procedure of a deep architecture unintuitive in the sense that it is difficult, if not impossible, to attribute the poor performance of a model to an improper design choice in a certain part or layer of the network. Also, it is hard to assess the quality of training in the hidden layers or interpret the effect of the learning algorithm on the model during training.
>
>     As we have argued in our reply to all reviewers regarding interpretability, the layer-wise learning approach makes it possible to debug each layer individually since we now have a metric against which we can evaluate the performance of each layer separately. And it also has clearer learning dynamics compared to end-to-end training schemes.]
>
>
> - Regarding the criticism that BP forces intermediate layers to correct for "mistakes" made by layers higher up: it seems your layer-wise algorithm attempts to learn the best possible representation in first layer, and then progresses to the next layer where it tries to correct for the potential error of the first layer and so on. In other words it seems that the errors of layers are propagated from first to last, instead of last to first as in BP, but are still being propagated in a sense. I do not immediately see why propagation forward should be preferable. Can you please further explain this point?
>
> [We are not sure that we understood this comment correctly. Could the reviewer please be more specific about which part of the paper this comment is referred to? We do not think we have made this particular criticism about BP in Section 4.1 or any part of the paper. It would be helpful if the reviewer can identify the claim in our paper that caused this confusion so that we can rephrase it in a clearer way. Many thanks in advance.]
>
> ---------------------------------------------
>
> - It is proven in the appendix (Lemma B.3) that under certain conditions stacking additional layers never leads to degradation of training loss. Can you please clarify whether additional layers can be helpful even in the case where previous layers already succeeded in learning the optimal representation?
>
> [If by succeeding in learning the optimal representaion, the reviewer meant achieving zero loss, then Lemma B.3 would only give a guarantee that we can stack on more layers and still have the resulting model achieve zero loss at the latest layer added. In other words, stacking more layers will certainly not help with further reducing training loss in that case.
>
> However, in practice, although identical training losses were achieved sometimes while experimenting with models of different depth, many benefits were observed by choosing the deeper model in this case. For example, the upper layers usually converge significantly faster to a smaller or identical training loss compared to the lower layers and hence for example, a three-layer kMLP may take fewer total iterations to reach a certain training loss than a two-layer one. Also, as we have argued in Appendix B.5, upper layers usually have a certain degree of robustness to bad kernel parameterization, hence, making them easier to fine-tune.]
>
> ---------------------------------------------
>
> - (Sec 4.1) Layer-wise vs. network-wise optimality: I find the claim that BP-based learner is not aware of the network-wise optimality confusing. BP explicitly optimises for network-wise optimality and the relative contribution to the network-wise error of each weight is propagated accordingly. I suppose my confusion stems from lack of a clear description of what defines a learner "aware" or "blind" to network-wise optimality.
>
> [We do not recall having made the claim that BP is not aware of the network-wise optimality. We certainly agree with the reviewer. Note that the fundamental difficulties in Section 4.1 are for layer-wise learning only. Could the reviewer please be more specific about which sentence/part in Section 4.1 caused this confusion? We will rephrase accordingly.
>
> To clarify further, a learner is "aware" of network-wise optimality for each layer when it is an end-to-end learner since in that case, as the reviewer has pointed out, each weight update would be toward minimizing the loss for the entire network. In contrast, a learner is "blind" to network-wise optimality when it works in a layer-wise fashion, optimizing a loss that is not the loss for the entire network one layer at a time.]

---

> ### Author Response · Authors · 2018-11-08
> **Reply to Reviewer 1 [3/6]**
>
>
> - In general, I am not convinced layer-wise optimality is a useful criterion when what we want to achieve is network-wise optimality. As you show in the appendix, if layer-wise optimality is achieved then it implies network-wise optimality; however, layer-wise optimality is only a sufficient condition and likely not a necessary one (except for the simplified scenario studied in B.3).
>
> [Edit (Nov. 26): The proposed layer-wise learning algorithm learns network-wise optimality at each layer given that the regularization coefficient is chosen properly. Section 4.3 has been revised to justify this claim. Please refer to the newest manuscript for details.]
>
> - It is thus not clear to me why layer-wise training would always be preferable to network-wise training (e.g. using BP) especially because its greedy nature might intuitively prevent learning of hierarchical representations which are commonly claimed to be key to the success of neural networks. Can you please clarify?
>
> [Some of the benefits of layer-wise learning as compared to BP have been presented in our response to the reviewer's first comment in the DETAILED COMMENTS section. Please see above for details.
>
> Regarding learning hierarchical representations, we think that even if the learning algorithm is greedy, the representations are still built on top of each other and hence, they can still be hierarchical.
>
> Moreover, we think that it is still an open question whether BP in fact also implicitly learns a deep NN in a greedy, bottom-up fashion. Some empirical results seem to suggest that the answer is positive, i.e., hierarchical representations learned with BP might have also been learned implicitly in a layer-by-layer manner [1].]
>
> ---------------------------------------------
>
> - (Sec 4.2) I think it would be beneficial to state in the introduction that the "risk" is with respect to the hinge loss which is common in the SVM/kernel literature but much less in the deep learning literature and thus could surprise a few people when they reach this point.
>
> [We have updated the manuscript according to the reviewer's comment and we thank the reviewer for bringing this potential confusion into our attention.]
>
> -------------------------------
> FURTHER QUESTIONS:
> -------------------------------
>
> - From Lemma 4.3, it seems that the derived representation is only optimal with respect to the **upper bound** on the empirical risk (which for \tau >= 2 will be an upper bound on the population risk). I got slightly confused at this point as my interpretation of the previous text was that the representation is optimal with respect to the population risk itself. Does the upper bound have the same set of optima? Please clarify.

---

> ### Author Response · Authors · 2018-11-08
> **Reply to Reviewer 1 [4/6]**
>
>
> [The reviewer is absolutely correct. For \tau >= 4\sqrt{c/N} in Lemma 4.3 (\tau >= \frac{8L^(l)Cd_{l-1}}{\max (|c|, |a|)} \sqrt{c/N} in Lemma 4.5), the representation characterized by Lemma 4.3 (Lemma 4.5) is optimal w.r.t. an upper bound on the population risk. Note that N is the size of the training set and hence, for reasonably-sized datasets, these two conditions are relatively mild.
>
> The true population risk is almost never computable under machine learning settings since we do not make distributional assumptions. For this reason, we believe that it is commonly accepted and also useful to have optimality w.r.t. a bound on the population risk instead of the true population risk. In fact, optimality of this kind justifies many commonly used learning paradigms. For example, for NN in classification, the standard minimization of an empirical risk and a regularization term is considered to be optimal w.r.t. an upper bound on the true risk [2][3].
>
> With that said, we do think that it can be beneficial to investigate the possibility of tightening the bound or deriving new bounds using different techniques, as this may shed light on new optimal hidden representations and hence also new layer-wise learning algorithms. We leave this as future work.]
>
> ---------------------------------------------
>
> - (p.5) There are two assumptions that I find somewhat restrictive. Just before Lemma 4.3 you assume that the number of points in each class must be the same. Can you comment on whether you expect the same representation to be optimal for classification problems with significantly imbalanced number of samples per class?
>
> [We thank the reviewer for pointing this out. This assumption has been removed in the newest manuscript and we have made changes to the proof to justify the removal. Please refer to page 17 for details.]
>
> - The second assumption is after Lemma 4.4 where you state that the stationary kernel k^{l-1} should attain its infimum for all x, y s.t. || x - y || greater than some threshold. This does not hold for many of the popular kernels like RBF, Matern, or inverse multiquadric. Do you think this assumption can be relaxed?
>
> [We agree with the reviewer that this assumption is somewhat restrictive. Unfortunately, this assumption cannot be removed in our current proof of the lemma. We are actively working towards improving or completely removing this assumption. Nevertheless, we think that for kernels with light tails such as the RBF kernel, the value would decay quickly away from the origin (for RBF, the decay is exponentially fast). Hence in practice, we expect that this assumption would not be too far away from reality. In all of our experiments, we have used RBF kernels with varied kernel widths. And we have pointed out that RBF kernels do not strictly satisfy this assumption in Section 7.]
>
> ---------------------------------------------
>
> - (p.5) Choice of the dissimilarity measure for G: Can you provide more intuition about why you selected L^1 distance and whether you would expect different results with L^2 or other common metrics?
>
> [The choice of dissimilarity measure is somewhat arbitrary. We do not have a specific reason as to why L^1 distance should be favored over L^2 distance or alignment. And we chose L^1 distance just so that we could obtain a concrete result (Lemma 4.5). Although, we should point out that the proof for Lemma 4.5 used the fact that the loss is L^1 distance. We are currently working on producing equivalent results when the loss is L^2 distance or alignment. In practice, however, we have not noticed any significant performance difference among using different loss functions for the hidden layers in our experiments. Hence we expect theoretical results equivalent to Lemma 4.5 to continue to hold for different losses.]
>
> ---------------------------------------------
>
> - (Sec 4.3) Can you please provide more details about the relation of the proposed objective (\hat{R}(F) + \tau max_j ||f_j||_H) to Lemmas 4.3 and 4.5 where the optimal representation was derived for functions that optimise an upper bound in terms of Gaussian complexity (e.g. is the representation that minimises risk w.r.t. the Gaussian bound also optimal with respect to functions that optimise this objective)?
>
> [The bounds in Lemma 4.3 and 4.5 using Gaussian complexity should be combined with the bound on Gaussian complexity in Theorem 4.2. This would give the objective \hat{R}_l(f^(l)) + \tau ||f^(l)||_{H_l} for Lemma 4.3 and \hat{R}_{l-1}(F^(l-1)) + \tau max_j ||f_j^(l-1)||_{H_{l-1}} for Lemma 4.5. We have updated the statements of Lemma 4.3 and 4.5 to clarify and we thank the reviewer for pointing this out.
>
> Note that a bound containing Gaussian complexity cannot be computed or used as a loss function since Gaussian complexity is not computable in practice (it involves computing the expectation of i.i.d. random variables of an unknown distribution).]

---

> ### Author Response · Authors · 2018-11-08
> **Reply to Reviewer 1 [5/6]**
>
>
> ---------------------
> EXPERIMENTS:
> ---------------------
>
> - I would appreciate addition of some standard baselines, like MLP combined with dropout or batch normalisation, and optimised with RMSProp (or similar). These would greatly help with assessing competitiveness with current SOTA results.
>
> - It would be nice to see the relative contribution of the two main components of the paper. Specifically, an experiment which would evaluate empirical performance of KNs optimised by some form of gradient descent vs. by your layer-wise training rule would be very insightful.
>
> [Edit (Nov. 26): We have added all the requested comparisons. Please refer to our summary of all changes made to the manuscript for details.]
>
> -----------
> OTHER:
> -----------
>
> - (p.2, 1st par in Sec 2) [minor] You state "a kernel machine is a universal function approximator". I suppose that might be true for a certain class of kernels but not in general?! Please clarify.
>
> [Indeed, this is true for the certain kernels only [4][5]. We have updated the manuscript to be more specific about this and we thank the reviewer for pointing it out.]
>
> ---------------------------------------------
>
> - (p.2, 3rd par in Sec 2) [minor] Are you using a particular version of the representer theorem in the representation of f_j^(i) as linear combination of feature maps? Please clarify.
>
> [We are only using the most standard version of the representer theorem [6]. And we have updated the manuscript to clarify this.]
>
> ---------------------------------------------
>
> - (p.2, end of 1st par in Sec 3) L^(i) is defined as sup over X_i. It is not clear to me that this constant is necessarily finite and I suspect it will not be in general (it will for the RBF kernel (and most stationary kernels) used in experiments though). Finiteness of L^(i) is necessary for the bound in Eq. (2) to be non-vacuous. Please clarify.
>
> [It is indeed part of the assumption that L^(i) < \infty. And we have updated the manuscript accordingly. Fortunately, as the reviewer pointed out, this assumption is satisfied by most popular kernels.]
>
> ---------------------------------------------
>
> - (p.3, after 1st display in Sec 4.2.1) [minor] Missing dot after "that we wish to minimise".
>
> [Could the reviewer please clarify on this comment? We could not figure out what the reviewer meant by "missing dot". Thanks in advance.]
>
> - Next sentence states "**the** optimal F" (emphasis mine) -- I am sorry if I overlooked it, but I did not notice a proof that a solution exists and is unique, and am not familiar enough with the literature to immediately see the answer. Perhaps a footnote clarifying the statement would help.
>
> [Without further assumption on the loss and the hypothesis space, such a global minimum may not be found in practice and also may not be unique (multiple risk-equivalent solutions may exist). Here we are only referring to the best F the learner can find that minimizes R_l. And it is not important whether that F is "the best" or if "the best" exists at all. We have updated the draft to clarify.]
>
> ---------------------------------------------
>
> - (p.4, 1st par in Sec 4) You say "A generalisation to regression is reserved for future work". I did not expect that based on the first few pages. On high-level, it seems that generalisation to regression need not be trivial as, for example, the optimal representation derived in Lemma 4.3 and Lemma 4.5 explicitly relies on the classification nature of the problem. Can you comment on expected difficulty of extension to regression?
>
> [Undoubtedly, generalization of our layer-wise learning algorithm to regression is a nontrivial task. At this point, we only have some very early ideas as to how to proceed. The main difficulty, in our attempt to come up with an equivalent theory for regression, is that the domain is now uncountable instead of a finite set.]
>
> - Possibly state in the introduction that only classification is considered in this paper.
>
> [We have now stated in the Abstract and also in the Introduction that our theoretical results regarding optimal hidden representations are only for classification and are proven only under certain losses.]
>
> ---------------------------------------------

---

> ### Author Response · Authors · 2018-11-08
> **Reply to Reviewer 1 [6/6]**
>
>
> - (p.7, 1st par in Sec 6) [related] "However they did not extend the idea to any **arbitrary** NN" (emphasis mine). Can you please be more specific here?
>
> [We meant that the early works that tried to "kernelize" NNs considered only certain NN architectures. As we have listed in Section 6, the most general work in this regard proposed only kMLP and the KN equivalent of CNN [7]. In contrast, we proposed that one can convert any NN to an equivalent KN by the simple procedure described in Section 2.]
>
> ---------------------------------------------
>
> - (p.5-6) [minor] Last sentence in Lemmas 4.3 and 4.5 is slightly confusing. Can you rephrase please?
>
> [Edit (Nov. 26): These two lemmas have been rephrased in the newest version of the paper. We thank the reviewer for pointing this out.]
>
> ---------------------------------------------
>
> - (p.6) [minor] You say "the learned decision boundary would generalise better to unseen data". Can you please clarify the last sentence (e.g. being more precise about the meaning of the word "simple" in the same sentence) and provide reference for why this is necessarily the case?
>
> [That the learned decision boundary would generalize better to unseen data follows from the fact that the optimal hidden representation at the last layer is so defined such that the bound on the expected classification error of the classifier is minimized. Hence if the l-1^th layer learns a representation that is close to this optimal one, it is of course reasonable to expect the classifier to generalize better to unseen data (sampled from the same distribution as that of the training set).
>
> By "simple", we meant that examples from distinct classes would be far apart in the RKHS and those in the same class would be close. This observation is justified in the proof of Lemma 4.3. And intuitively, this representation is "simple" to classify.]
>
>
>
> [1] Raghu, M., Gilmer, J., Yosinski, J., & Sohl-Dickstein, J. (2017). Svcca: Singular vector canonical correlation analysis for deep learning dynamics and interpretability. In Advances in Neural Information Processing Systems (pp. 6076-6085).
>
> [2] Bartlett, P. L. (1997). For valid generalization the size of the weights is more important than the size of the network. In Advances in neural information processing systems (pp. 134-140).
>
> [3] Bartlett, P. L., & Mendelson, S. (2002). Rademacher and Gaussian complexities: Risk bounds and structural results. Journal of Machine Learning Research, 3(Nov), 463-482.
>
> [4] Park, J., & Sandberg, I. W. (1991). Universal approximation using radial-basis-function networks. Neural computation, 3(2), 246-257.
>
> [5] Micchelli, C. A., Xu, Y., & Zhang, H. (2006). Universal kernels. Journal of Machine Learning Research, 7(Dec), 2651-2667.
>
> [6] Schölkopf, B., Herbrich, R., & Smola, A. J. (2001, July). A generalized representer theorem. In International conference on computational learning theory (pp. 416-426). Springer, Berlin, Heidelberg.
>
> [7] Zhang, S., Li, J., Xie, P., Zhang, Y., Shao, M., Zhou, H., & Yan, M. (2017). Stacked Kernel Network. arXiv preprint arXiv:1711.09219.

---

> ### Author Response · Authors · 2018-11-12
> **New manuscript has been uploaded**
>
>
> This post contained a description of updates in a past version of the paper. For clarity, all the changes we have made during the rebuttal period to the manuscript are now summarized in the newest reply to all reviewers. Please refer to that for details.

---

> ### Author Response · Authors · 2018-11-15
> **New manuscript with more experimental results has been uploaded**
>
>
> This post contained a description of updates in a past version of the paper. For clarity, all the changes we have made during the rebuttal period to the manuscript are now summarized in the newest reply to all reviewers. Please refer to that for details.

---

> ### Author Response · Authors · 2018-11-19
> **New manuscript has been uploaded**
>
>
> This post contained a description of updates in a past version of the paper. For clarity, all the changes we have made during the rebuttal period to the manuscript are now summarized in the newest reply to all reviewers. Please refer to that for details.

---

> ### Author Response · Authors · 2018-11-26
> **New results on Fashion-MNIST**
>
>
> This post contained a description of updates in a past version of the paper. For clarity, all the changes we have made during the rebuttal period to the manuscript are now summarized in the newest reply to all reviewers. Please refer to that for details.

---

> ### Author Response · Authors · 2018-12-04
> **Thank you!**
>
>
> Dear Reviewer 1,
>
> Thank you so much for increasing the score. We couldn't have made those improvements without your very helpful review. Thanks again for reviewing our paper.

---

### Official Review · AnonReviewer2 · 2018-11-03
**Good paper, some things are oversold**

**Rating:** 6
**Confidence:** 4

**Review:**

This paper attempts to learn layers of NNs greedily one at a time by using kernel machines as nodes instead of standard nonlinearities. The paper is well-written and was an interesting read, despite being notation heavy.

I think the interpretability claims have some merits but are over-stated. Furthermore, the expressive power of universal approximation through kernels holds only asymptotically. So I am not sure if the authors can claim equivalence in expressive powers to more traditional NNs theoretically. I have some additional questions about the paper, and I am reserving my recommendation on this paper till the authors answer them.

1) Since individual node is simply a hyperplane in the induced kernel space, why not just specify the cost function as the risk + \tau * norm(weights) ?  What is the benefit of explicitly talking about gaussian complexities and delineating Theorem 4.2 when the same can be achieved by writing a much simpler form? Lemmas 4.4 and 4.5 should be straightforward extensions too if just used in this form since Lemma C.1 follows easily, and again could be simplified a lot by just using the regularized cost function. Am I missing something here?


2) Lemma 4.3 assumes separability (since c should be > a for \tau to be positive) of classes, and also balanced classes (since number of positives = number of negatives). Why are these assumptions reasonable ? I understand that the empirical evaluation presented do justify the methodology, but I am wondering if based on these assumptions the theoretical results are of any use in the way they are currently presented.

Minor :
Below Def 4.1  "to a standard normal distribution " should be "according to P".
Some typos, please proof read e.g. spelling error "represnetation ".

---

> ### Author Response · Authors · 2018-11-08
> **Reply to Reviewer 2 [1/2]**
>
>
> Firstly, we would like to thank Reviewer 2 for the insightful review. We have found the comments and questions really helpful and we now address them in details. We do hope that Reviewer 2 finds our response satisfying.
>
> Comments from the reviewer are listed first with each preceded by a dash. Our replies are put in brackets.
>
> -----------------
> COMMENTS:
> -----------------
>
> - I think the interpretability claims have some merits but are over-stated.
>
> [We are thankful that the reviewer brought up this important issue. Please see our reply to all reviewers for our response.]
>
> ---------------------------------------------
>
> - Furthermore, the expressive power of universal approximation through kernels holds only asymptotically. So I am not sure if the authors can claim equivalence in expressive powers to more traditional NNs theoretically.
>
> [Generally, we think that the issue of expressive power is a highly abstract one and it is usually difficult to argue which model possesses stronger expressive power in very concrete terms. Although, we do agree with the reviewer that the universal approximation property of kernel machines holds only when we do not limit the number and the positions of its centroids [1]. Nevertheless, to the best of our knowledge, all classical universal approximation results for NNs are also asymptotical results [2][3][4].
>
> Moreover, intuitively, a single node in a KN (a kernel machine) is already (asymptotically) a universal function approximator. In contrast, it takes at least two layers of NN to be (asymptotically) a universal function approximator.
>
> Further, in terms of some complexity measures such as Gaussian complexity, Lemma B.2 in our paper seems to show that the model complexity of kMLP is comparable to that of MLP [5]. And they also scale in a similar way in the depth and width of the network.
>
> Combining the arguments above, we expect KN to be at least comparable to NN in terms of expressive power, which is corroborated by our experimental results in the paper.]

---

> ### Author Response · Authors · 2018-11-08
> **Reply to Reviewer 2 [2/2]**
>
>
> -----------------
> QUESTIONS:
> -----------------
>
> - 1) Since individual node is simply a hyperplane in the induced kernel space, why not just specify the cost function as the risk + \tau norm(weights) ?  What is the benefit of explicitly talking about gaussian complexities and delineating Theorem 4.2 when the same can be achieved by writing a much simpler form?
>
> [The fact that minimizing empirical loss + \tau norm (weights) effectively minimizes the true (expected) risk is ultimately justified for NN by bounds analogous to that in Theorem 4.2 [6][7][8]. And since KN is structurally different from NN, we wanted to be more rigorous by deriving similar bounds for our novel architecture and justifying the effectiveness of the layer-wise training algorithm from first principles. In other words, we wanted to prove that the learning algorithm we designed for KN would actually minimize the true risk and hence guarantee generalization to unseen data.
>
> Despite that, as the reviewer has pointed out, each node in a KN is a hyperplane in an RKHS, it is not entirely clear to us how the above result should follow easily as, after all, the hyperplane is in an RKHS instead of the input space.]
>
> - Lemmas 4.4 and 4.5 should be straightforward extensions too if just used in this form since Lemma C.1 follows easily, and again could be simplified a lot by just using the regularized cost function. Am I missing something here?
>
> [The difficulty in going from Theorem 4.2 + Lemma 4.3 to Lemma 4.4 + Lemma 4.5 is that the losses are different and so are the dimensions of the layers. These two factors eventually required a different bound (Lemma 4.4) and also a new proof of optimality for the so-defined optimal representation in Lemma 4.3.]
>
> ---------------------------------------------
>
> - 2) Lemma 4.3 assumes separability (since c should be > a for \tau to be positive) of classes, and also balanced classes (since number of positives = number of negatives). Why are these assumptions reasonable ? I understand that the empirical evaluation presented do justify the methodology, but I am wondering if based on these assumptions the theoretical results are of any use in the way they are currently presented.
>
> [About the first assumption, recall from Section 3 that a := min_{x, y} k(x, y) and c := max_{x, y} k(x, y). Hence, unless the kernel chosen is a constant (which is of course not of interest in practice), a < c by construction. So we are really making no assumption here. Moreover, just to clarify, neither Lemma 4.3 nor Lemma 4.5 requires a separability assumption.
>
> In terms of the assumption that the classes are balanced, we have been working on it since the initial submission and it turns out that this assumption is not needed at all. We have updated the proof. Please see the new proof on page 16 for details. We thank the reviewer for bringing this up.]
>
> ----------
> MINOR:
> ----------
>
> - Below Def 4.1  "to a standard normal distribution" should be "according to P".
>
> [Here, we refer to the sequence g_1, ..., g_N as the sequence that is being ``fitted''. And the g_n's are i.i.d. standard normal random variables. This interpretation was taken directly from [7]. We have updated the manuscript to clarify.]
>
> ---------------------------------------------
>
> - Some typos
>
> [We have corrected the typos in the newly uploaded version and we thank the reviewer for brining them into our attention.]
>
>
>
> [1] Park, J., & Sandberg, I. W. (1991). Universal approximation using radial-basis-function networks. Neural computation, 3(2), 246-257.
>
> [2] Cybenko, G. (1989). Approximation by superpositions of a sigmoidal function. Mathematics of control, signals and systems, 2(4), 303-314.
>
> [3] Funahashi, K. I. (1989). On the approximate realization of continuous mappings by neural networks. Neural networks, 2(3), 183-192.
>
> [4] Barron, A. R. (1993). Universal approximation bounds for superpositions of a sigmoidal function. IEEE Transactions on Information theory, 39(3), 930-945.
>
> [5] Sun, S., Chen, W., Wang, L., Liu, X., & Liu, T. Y. (2016, February). On the Depth of Deep Neural Networks: A Theoretical View. In AAAI (pp. 2066-2072).
>
> [6] Bartlett, P. L. (1997). For valid generalization the size of the weights is more important than the size of the network. In Advances in neural information processing systems (pp. 134-140).
>
> [7] Bartlett, P. L., & Mendelson, S. (2002). Rademacher and Gaussian complexities: Risk bounds and structural results. Journal of Machine Learning Research, 3(Nov), 463-482.
>
> [8] Shalev-Shwartz, S., & Ben-David, S. (2014). Understanding machine learning: From theory to algorithms. Cambridge university press.

---

> ### Author Response · Authors · 2018-11-12
> **New manuscript has been uploaded**
>
>
> This post contained a description of updates in a past version of the paper. For clarity, all the changes we have made during the rebuttal period to the manuscript are now summarized in the newest reply to all reviewers. Please refer to that for details.

---

> ### Author Response · Authors · 2018-11-15
> **New manuscript with more experimental results has been uploaded**
>
>
> This post contained a description of updates in a past version of the paper. For clarity, all the changes we have made during the rebuttal period to the manuscript are now summarized in the newest reply to all reviewers. Please refer to that for details.

---

> ### Author Response · Authors · 2018-11-19
> **New manuscript has been uploaded**
>
>
> This post contained a description of updates in a past version of the paper. For clarity, all the changes we have made during the rebuttal period to the manuscript are now summarized in the newest reply to all reviewers. Please refer to that for details.

---

> ### Author Response · Authors · 2018-11-26
> **New results on Fashion-MNIST**
>
>
> This post contained a description of updates in a past version of the paper. For clarity, all the changes we have made during the rebuttal period to the manuscript are now summarized in the newest reply to all reviewers. Please refer to that for details.

---

> ### Author Response · Authors · 2018-12-19
> **Your thoughts on our response?**
>
>
> Dear Reviewer 2,
>
> Hello! We really hope we have fully addressed your concerns in our earlier reply and it would be great if you could give us some feedback. In particular, if you think we have adequately addressed your two major technical concerns, could you please reconsider your rating? Thank you very much!
>
> Best regards,
> Paper 1364 authors

---

### Author Response · Authors · 2018-11-08
**Note on interpretability**


CONCERN ON INTERPRETABILITY:

All three reviewers pointed out that it is somewhat imprecise to claim that KN or the layerwise learning paradigm is more interpretable than NN trained with BP. We appreciate the reviewers for bringing up this important issue and we will try our best to clarify our claim.

First of all, we think that KN is more interpretable in the sense that thanks to the use of kernel function, the model can be embedded in an inner product space in which it is linear. The inner product space provides constructions to interpret learning geometrically and the model being linear makes it easy to visualize and to work with. This enables us to reduce rather abstract problems such as what is the best hidden representation at a given layer to geometric ones. Our derivation of the optimal hidden representations essentially utilized this nice property of KN. Also, this provided a straightforward geometric interpretation of the learning dynamics in greedily-trained KN, as we have discussed in page 5 and page 6.

For NN, the nonlinearities lack such a natural mathematical construction in which we can easily talk about geometric concepts such as distance, angle, etc. Note that although we can embed the hidden activations of an NN in some proper Euclidean space, the model is still not linear and hence is difficult to deal with in that space, defeating the purpose of such an embedding. Moreover, in contrast to KN learned layer-by-layer, the interpretation of the learning dynamics of NN learned with BP remains a challenging theoretical problem. Interestingly, the most notable work along this line of research showed (empirically) that the learning dynamics in NN seem to agree with our theory for KN [1].

Moreover, the design process of a KN is now more transparent and intuitive with layer-wise training. This is because by construction of the layer-wise learning algorithm, the quality of hidden representations can be evaluated explicitly at each layer, which provides more information to the user and makes it possible to trace the bad performance of the overall model to a certain layer and debug the layers individually.

Although, we are not claiming that KN is as transparent as a simple linear model in which the contribution of each input feature to the output can be directly identified. We agree that in that sense, KN and NN are both difficult, if not impossible, to interpret. We agree that this is perhaps the more commonly used definition for model interpretability in machine learning. And we have changed the title of the paper as well as the parts in which interpretability of KN is discussed to avoid any confusion. The changes are reflected in the newest manuscript.



[1] Raghu, M., Gilmer, J., Yosinski, J., & Sohl-Dickstein, J. (2017). Svcca: Singular vector canonical correlation analysis for deep learning dynamics and interpretability. In Advances in Neural Information Processing Systems (pp. 6076-6085).

---

### Author Response · Authors · 2018-11-27
**Summary of changes made to the manuscript**


Dear reviewers,

Here is a summary of all the major changes we have made to our paper according to your comments and requests. We do hope that we have fully addressed your concerns.

1. New experimental results. As requested by Reviewer 1, we have added experimental results to better complement our theory. These include

    1) Results of greedy kMLPs on two new datasets (standard MNIST and Fashion-MNIST [1]).
    2) New standard MLP baselines including MLPs trained with SGD, Adam, RMSProp+batch normalization and RMSProp+dropout on all datasets.
    3) Comparisons of greedy kMLPs and kMLPs trained with standard BP on MNIST.

In these experiments, the greedy kMLPs compared favorably with the MLPs even though no advanced training techniques such as batch normalization or dropout was used for the former. Also, for both the single-hidden-layer and the two-hidden-layer kMLPs, the layer-wise algorithm consistently outperformed BP.

2. As pointed out by Reviewer 1, Section 4.3 has been revised to emphasize that the proposed layer-wise algorithm learns network-wise optimality at each layer, justifying the optimality and practicality of the algorithm.

3.  We have reformulated the two key lemmas (Lemma 4.3 and Lemma 4.5), as requested by Reviewer 1. We think this new formulation is clearer than the previous one.

4.  We have rewritten the definition of f^(i)_j in par. 3, Section 2, since Reviewer 3 mentioned that the original one was unclear. Also, as requested by Reviewer 1, we have justified expanding the kernel machine on the training sample using the generalized representation theorem [2]. This theorem directly applies in the layer-wise setting.

5. As pointed out by Reviewer 1 and 2, the balanced class assumption of Lemma 4.3 has now been removed. The proof has also been updated to justify the removal.



[1] Xiao, H., Rasul, K., & Vollgraf, R. (2017). Fashion-mnist: a novel image dataset for benchmarking machine learning algorithms. arXiv preprint arXiv:1708.07747.

[2] Schölkopf, B., Herbrich, R., & Smola, A. J. (2001, July). A generalized representer theorem. In International conference on computational learning theory (pp. 416-426). Springer, Berlin, Heidelberg.

---

### Meta-Review · Area_Chair1 · 2018-12-18

**Confidence:** 5
**Recommendation:** Reject

**Metareview:**

The reviewers mostly raised two concerns regarding the paper: a) why this algorithm is more interpretability than BP (which is just gradient descent); b) the exposition of the paper is somewhat confusing at various places; c) the lack of large-scale experiment results to show this is practically relevant. In the AC's opinion, a principled kernel-based approach can be counted as interpretable, and there the AC would support the paper if a) is the only concern. However, c) seems to be a serious concern since the paper doesn't seem to have experiments beyond fashion MNIST (e.g., CIFAR is pretty easy to train these days) and doesn't have experiments with convolutional models. Based on c), the AC decided that the paper is not quite ready for acceptance.